# Learning-Augmented Online Minimization with Dual Predictions

**Christian Coester** [* 1]  **Alexa Tudose** [* 1]  **Alexander Turoczy** [* 1]

## Abstract

We present learning-augmented algorithms for two general classes of online minimization problems: metrical task systems and laminar set cover. Both algorithms achieve improved theoretical guarantees using machine-learned predictions of an optimal solution to the dual linear program. Unlike optimal primal solutions, which can change drastically under tiny instance perturbations, these dual solutions are much more stable, which ensures the existence of good (and learnable) predictions for families of similar instances. While previous work has used dual predictions in offline settings and for online maximization problems, our algorithms are, to the best of our knowledge, the first demonstration that such dual predictions can be effective for online minimization. Our theoretical results are complemented by experiments on the $k$-server problem and the parking permit problem.

## 1. Introduction

Many decision-making problems arise in settings where inputs arrive over time (e.g., as a sequence of requests) and decisions must be made without knowing the future. Such problems are commonly modeled as *online* optimization problems, where an algorithm must act irrevocably as each request arrives. The classical analysis of online algorithms assumes that the input is completely unknown and may even be chosen adversarially. Online algorithms are then evaluated by their *competitive ratio*, defined as the worst-case ratio between the cost of the online algorithm and the best solution in hindsight. This viewpoint has led to a rich theory, and to strong impossibility results, but it can feel misaligned with practice, where similar instances of the

*same* optimization task are solved day after day. Modern systems often have access to historical data, forecasts, or learned models that can try and predict aspects of the future.

Motivated by this, the field of *learning-augmented algorithms* (a.k.a. *algorithms with predictions*) has emerged in recent years (Lykouris & Vassilvitskii, 2021; Purohit et al., 2018; Mitzenmacher & Vassilvitskii, 2020). Here, an algorithm's input is augmented with predictions about the instance; the goal is to design algorithms that benefit from predictions if these are reasonably accurate, while still satisfying classical worst-case bounds even when predictions are highly erroneous.

This paradigm has spurred progress across many online problems, including paging/caching, rent-or-buy problems, scheduling, and graph and metric optimization. Learning-augmented algorithms have also been developed in many other areas of research, such as to improve running times, data structures, approximation algorithms, streaming algorithms, and in game theory. We refer to Lindermayr & Megow (2026) for an extensive repository of research on learning-augmented algorithms.

**What to Predict?** A central design choice is to determine what information should be predicted. Common approaches for online minimization problems have been *event predictions* (e.g., time or location of future requests) and *action predictions* (i.e., suggested actions that the algorithm should take, intended to represent a near-optimal primal solution to the optimization problem). For example in caching, event predictions in the form of next request times for each page have been used successfully (Lykouris & Vassilvitskii, 2021; Rohatgi, 2020; Wei, 2020).

However, these types of predictions come with some drawbacks. One drawback of event predictions for caching is that they are *instable*: A single change in an otherwise perfectly predicted sequence can lead to an unbounded prediction error (because an anticipated event may be delayed indefinitely). A consequence of this instability is that a *single* incorrect prediction can completely deteriorate the performance of the state-of-the-art learning-augmented algorithm for the problem, making it perform as poorly as algorithms *without* predictions. We show an example of such behavior in Appendix A.2. Notably, this does not contradict the

---

*Equal contribution [1]University of Oxford, United Kingdom. Correspondence to: Christian Coester <christian.coester@cs.ox.ac.uk>, Alexa Tudose <maria-alexa.tudose@cs.ox.ac.uk>, Alexander Turoczy <alexandertur-oczy@gmail.com>.

*Proceedings of the 43rd International Conference on Machine Learning*, Seoul, South Korea. PMLR 306, 2026. Copyright 2026 by the author(s).

algorithm's *smoothness* property (i.e., competitive ratio degrades gracefully as a function of prediction error), which may seem surprising as smoothness is motivated by the desire to prevent such scenarios. But smoothness only bounds the algorithm's performance in terms of prediction error, but it fails to ensure that prediction error is stable under small instance changes.

Beyond their instability, another downside of these event predictions for caching is that already for the slightly more general problem of weighted caching (and consequently further generalizations like $k$-server and metrical task systems), even *exact knowledge* of the next request time of each page/location was shown to be insufficient for improved algorithmic guarantees (Bansal et al., 2022; Jiang et al., 2022).

Action predictions suffer from a similar problem of instability: optimal primal solutions can be extremely fragile. Small perturbations to an instance can completely change the structure of an optimal primal assignment (see Appendix A.1). As a consequence, it is often hard to justify why action predictions would be available (i.e., why they are *learnable*). While the usefulness of action predictions is intuitively evident (if you get good advice of what to do, just do it), action predictions shift an enormous burden onto the predictor, as it is a strong assumption that high-quality action advice is given.

We advocate a different prediction interface: *learn the dual*. Concretely, we study online minimization problems that admit natural linear programming (LP) formulations and design learning-augmented algorithms whose predictions are assignments to variables of the corresponding dual LP. Our thesis is that for a wide range of problems, with an appropriate LP formulation and prediction-error notion, dual predictions can simultaneously satisfy three desiderata:

1. **Stability:** small changes in the instance cause only small prediction error;

2. **Usefulness:** small prediction error implies near-optimal performance, and the guarantee degrades smoothly with the error (and can be bounded robustly even for very large error);

3. **Learnability:** the prediction target can be learned from a polynomial number of samples from a distribution.

At a high level, stability and learnability play complementary roles: while learnability ensures that the best prediction for a given distribution can be identified with small sample complexity, stability justifies that a good prediction exists in the first place for a family of similar instances.

We instantiate this approach in two fairly general online minimization domains.

**Laminar Set Cover.** Set cover is a central problem in online algorithms, and many competitive algorithms for seemingly different problems are derived via equivalent reformulations as (set) covering problems (Buchbinder & Naor, 2009). We study a special case where the set family is laminar, a constraint that is natural in "interval-like" covering settings. This captures problems such as ski rental (Phillips & Westbrook, 1999), dynamic power management (Antoniadis et al., 2021b), the parking permit problem (PPP) (Meyerson, 2005), the multi parking permit problem (de Lima et al., 2017), the Steiner leasing problem (Meyerson, 2005), and sum-radii clustering in ultrametrics (a.k.a. HSTs) (Fotakis & Koutris, 2014). The best achievable competitive ratio without predictions depends on the type of problem within this class of laminar set cover. In general laminar set cover, a lower bound of $\Omega(\log f)$ holds by the reduction from the parking permit problem (Meyerson, 2005), where $f$ is the maximum number of sets that can contain an element. On the other hand, this bound is also a valid upper bound for arbitrary laminar set cover instances due to the matching bound for general (non-laminar) fractional set cover (Alon et al., 2009) and a lossless rounding from fractional to randomized integral algorithms (Lemma 2.1).

**Metrical Task Systems (MTS).** MTS is a foundational online problem introduced by Borodin, Linial, and Saks (Borodin et al., 1992) and extensively studied since. It is a broad generalization of many online problems, including caching, $k$-server, convex function chasing, dynamic power management, and layered graph traversal, and with links to the experts problem in online learning (Blum & Burch, 2000). In an MTS instance, an algorithm moves in a metric space to service requests, and the goal is to minimize service and movement costs. In the classical setting without predictions, for any $n$-point metric, the competitive ratio of deterministic algorithms is exactly $2n - 1$ (Borodin et al., 1992) and for randomized algorithms it is between $\Omega(\log n)$ and $O(\log^2 n)$, where both the lower and upper bounds are tight for some metrics (Bubeck et al., 2021; Coester & Lee, 2022; Bubeck et al., 2023). Better bounds are achievable for special cases of MTS, as these correspond to restricting the class of cost functions.

## 1.1. Our Results

**Notation.** We specify some standard notation we will use throughout the paper. We use ALG to denote either a given algorithm or its cost on a given instance, and OPT for the optimal offline cost. ALG is $\mathcal{R}$-competitive if $\mathbb{E}[\text{ALG}] \leq \mathcal{R} \cdot \text{OPT}$ on any instance. We use $\eta$ to denote a measure of error for a given prediction, which is defined in a problem-specific manner. We also define the normalized $\bar{\eta} := \frac{\eta}{\text{OPT}}$.

We prove that dual predictions for both Laminar Set Cover and Metrical Task Systems fulfill the desired qualities of

usefulness, stability, and learnability. Our results are summarized in the theorems below.

**Theorem 1.1.** *(Informal).* *Suppose there is an $\mathcal{R}$-competitive online algorithm for a class of laminar set cover instances. Then predictions $\hat{\mathbf{y}}$ to an optimal dual solution satisfy*

(a) *(Usefulness) For any constant $\epsilon > 0$, there exists a learning-augmented algorithm which satisfies $\mathbb{E}[\text{ALG}] = (1 + \epsilon)\text{OPT} + O\left(\frac{\mathcal{R}\eta}{\epsilon}\right)$.*

(b) *(Stability) For problem instances $X_1$ and $X_2$, there exist optimal duals $\mathbf{y}_1^*, \mathbf{y}_2^*$ satisfying $\|\mathbf{y}_1^* - \mathbf{y}_2^*\|_1 = O(|X_1 \Delta X_2|)$, where $\Delta$ denotes symmetric difference.*

(c) *(Learnability) The predictions are learnable with polynomial sample complexity.*

(d) *(Culmination) With access to a polynomial number of i.i.d. samples $\boldsymbol{\sigma} \sim \mathcal{D}$, there exists an algorithm satisfying $\mathbb{E}_{\boldsymbol{\sigma} \sim \mathcal{D}}[\text{ALG}] = O\left(\mathbb{E}_{\boldsymbol{\sigma} \sim \mathcal{D}}[\text{OPT}] + \mathcal{R} \cdot \text{Var}(\boldsymbol{\sigma})\right).$*

**Theorem 1.2.** *(Informal). For MTS, predictions $\widehat{w}$ of an optimal dual solution satisfy*

(a) *(Usefulness) There exists an algorithm augmented with predictions $\widehat{w}$ which satisfies $\text{ALG} \leq \text{OPT} + \eta$.*

(b) *(Stability) Even if $\widehat{w}$ is not the optimal dual for the instance at hand, but rather for a nearby instance, the error in the prediction is small.*

(c) *(Learnability) The predictions are learnable with polynomial sample complexity.*

We prove usefulness and stability in Sections 2 and 3, and we defer the learnability results to the appendix. Additionally, we show in Appendix A.3 that unlike for the laminar version, dual predictions are not helpful for the standard LP relaxation of general set cover – there are worst-case instances that remain hard even when an optimal dual solution is known upfront.

**Robustness.** Our algorithms can be made robust against high prediction error by using standard algorithm combination techniques, which can be found in the appendix.

**Experiments.** Complementing our theoretical results, Section 4 presents a short experimental evaluation of our algorithms instantiated for the $k$-server problem and the parking permit problem. Using real-world data, we observe substantial performance benefits compared to traditional online algorithms.

## 1.2. Related Work

**Related Prediction Setups.** Dual variables and primal–dual reasoning are ubiquitous in online algorithms (Buchbinder & Naor, 2009). In learning-augmented algorithms, Bamas et al. (2020) presented a method to augment primal–dual algorithms using *primal* predictions (i.e., action predictions). Conceptually similar to our dual predictions is the *learned weights* method of Lattanzi et al. (2020) for scheduling with restricted assignment. Lavastida et al. (2021) show that these weights are stable (termed *instance-robust* there) and learnable. They do not, however, correspond to duals of the primal minimization problem. Li & Xian (2021) extend results from these works to scheduling on unrelated machines, using predictions consisting of both machine weights *and* dual variables. The dual variables are used here only to reduce from the unrelated machines setting to related machines restricted assignment, and the resulting instance is solved using the machine weight predictions similarly to Lattanzi et al. (2020). A recent line of work by Cohen and Panigrahi further generalizes these results to online allocation problems, and shows that the machine weights can be interpreted as dual variables of an *auxiliary convex program* of maximizing entropy subject to feasibility and near-optimality constraints (Cohen & Panigrahi, 2023; 2025). Our focus is different: we design algorithms whose predictions consist *purely* of dual variables of the underlying primal minimization LP itself (not of an auxiliary convex optimization problem). To the best of our knowledge, this viewpoint has not previously been developed as a general design principle for learning-augmented online minimization.

Dual predictions have appeared in other learning-augmented settings outside our scope. On the online *maximization* side, learning dual "shadow" prices from data is a classic idea in stochastic and random-order models, notably in AdWords and related allocation problems (Devanur & Hayes, 2009; Feldman et al., 2010; An et al., 2024). On the offline side, learned duals can be used to warm-start and accelerate primal–dual optimization, e.g. for matching (Dinitz et al., 2021; Chen et al., 2022).

**(Laminar) Set Cover.** The online set cover problem (Alon et al., 2009) gave rise to the powerful online primal-dual framework with its many applications to countless problems (Buchbinder & Naor, 2009). Learning-augmented treatments of online covering include (Bamas et al., 2020; Anand et al., 2022; Gupta et al., 2022; Grigorescu et al., 2022; Zeynali et al., 2021; Azar et al., 2023; Ameli et al., 2025). An important special case of *laminar* set cover is the ski rental problem, studied extensively in the learning-augmented literature, e.g., Purohit et al. (2018); Gollapudi & Panigrahi (2019); Wei & Zhang (2020); Angelopoulos et al. (2024); Antoniadis et al. (2021a); Diakonikolas et al.

(2021). The parking permit problem (Meyerson, 2005) is the seminal leasing problem in online algorithms. It generalizes ski rental and can be modeled, up to a constant factor, as laminar set cover. In the learning-augmented setting, Yaroslav & Alexander (2024) studied the special case of three permit types. A recent preprint by Ameli et al. (2025) highlights the parking permit problem as an application of their learning-augmented algorithm for covering problems. They achieve competitive ratios of $O(\log \eta)$ deterministically and $O(\log \log \eta)$ randomized, where $\eta$ is the $\ell_1$ prediction error. These guarantees are strongest in regimes of bounded time horizon; in long-duration instances, $\eta$ may grow proportionally with the time horizon, in which case the bound can be weaker than classical bounds that depend on the number of permits rather than the time horizon.

**Metrical Task Systems.** In the classical setting without predictions, MTS is now well understood: For any $n$-point metric, the competitive ratio is exactly $2n - 1$ (Borodin et al., 1992) for deterministic algorithms and $O(\log^2 n)$ for randomized algorithms (Bubeck et al., 2021; Coester & Lee, 2022; Bubeck et al., 2023). Better bounds are achieved for special cases of MTS: For example, $k$-server is $O(k)$-competitive (Koutsoupias & Papadimitriou, 1995), convex body chasing in a $d$-dimensional space is $O(d)$-competitive (Sellke, 2020; Argue et al., 2021), and width-$w$ layered graph traversal is $\Theta(w^2)$-competitive randomized (Bubeck et al., 2022; 2023) and $\Theta(2^w)$-competitive deterministically (Fiat et al., 1998; Coester & Tudose, 2026).

Learning-augmented MTS have been considered in the setting with action predictions: Antoniadis et al. (2023c) give algorithms with competitive ratio parametrized by the prediction error, Christianson et al. (2023) analyze tradeoffs between consistency (i.e., competitive ratio relative to the predicted solution) and robustness (i.e., competitive ratio relative to the offline optimum), Antoniadis et al. (2023b) and Cosa & Eliás (2025) consider scenarios with multiple action predictors, and Sadek & Eliás (2024) study algorithms that use action predictions parsimoniously. Special cases of MTS that were studied in the learning-augmented setting include paging/caching (Lykouris & Vassilvitskii, 2021; Rohatgi, 2020; Wei, 2020; Antoniadis et al., 2023c; Jiang et al., 2022; Bansal et al., 2022; Antoniadis et al., 2023a), $k$-server (Lindermayr et al., 2025; Christianson et al., 2023), convex function chasing (Christianson et al., 2022; Lechowicz et al., 2024), and dynamic power management (Antoniadis et al., 2021a).

## 2. Laminar Set Cover

In the set cover problem, we have a ground set $\mathcal{U} = \{u_1, ..., u_n\}$, and a set family $\mathcal{S} = \{S_1, ..., S_m\}$ with $S_j \subseteq \mathcal{U}$ and associated costs $c_{S_j} \geq 0$. An algorithm must

then find a cover $\mathcal{C} \subseteq \mathcal{S}$, satisfying $\bigcup_{S \in \mathcal{C}} S = \mathcal{U}$. The total cost an algorithm seeks to minimize is $c(\mathcal{C}) := \sum_{S \in \mathcal{C}} c_S$. In the *online* set cover problem, only a subset of $X \subseteq \mathcal{U}$ must be covered, where the elements in $X$ are revealed sequentially in an online fashion. We use a discrete time model, where at time $t$, a request $e_t \in X$ is revealed and must be covered by some set in $\mathcal{S}$. Additionally, we suppose $\mathcal{S}$ is a laminar set family, meaning that any two $S, S' \in \mathcal{S}$ are either disjoint, or one set is a subset of the other.

The assumption of laminarity crucially means that the sets in $\mathcal{S}$ form a natural hierarchy, where sets higher in the hierarchy contain all sets lower in the hierarchy. We assume without loss of generality that $\mathcal{U} \in \mathcal{S}$. We represent $\mathcal{S}$ as a rooted tree $\mathcal{T} = (\mathcal{S}, E)$, where the vertices are the sets of $\mathcal{S}$, arcs $a = (S, S')$ represent that $S \supseteq S'$, and $\mathcal{U}$ is the root.

We further assume without loss of generality that if $S \subsetneq S'$, then $c_S < c_{S'}$, as otherwise in any feasible solution we can replace any $S$ with $S'$ without increasing the cost.

An LP relaxation of online laminar set cover is given by

$$\begin{aligned} \min \quad & \sum_{S \in \mathcal{S}} c_S x_S \\ \text{s.t.} \quad & \sum_{S: e_t \in S} x_S \geq 1, \quad \forall e_t \\ & x_S \geq 0, \qquad \forall S \in \mathcal{S}. \end{aligned}$$

Variable $x_S$ denotes the fraction of set $S$ bought. In the online setting, constraints are revealed one by one and an algorithm may only increase $x_S$ over time. We call this the *fractional* version of the problem. It suffices to solve this version of the problem due to a simple online randomized rounding procedure for laminar set cover implicit in the work of Meyerson (2005).

**Lemma 2.1.** *There exists a randomized online rounding algorithm,* $\mathrm{ALG}_{int}$, *which given a deterministic online algorithm* $\mathrm{ALG}_{frac}$ *for the online fractional laminar set cover problem satisfies* $\mathbb{E}[\mathrm{ALG}_{int}] = O(\mathrm{ALG}_{frac})$.

The corresponding dual LP is given by

$$\begin{aligned} \max \quad & \sum_{e \in \mathcal{U}} y_e \\ \text{s.t.} \quad & \sum_{e \in S} y_e \leq c_S, \quad \forall S \in \mathcal{S} \\ & y_e \geq 0, \qquad \forall e \in \mathcal{U} \\ & y_e = 0, \qquad \forall e \notin X. \end{aligned}$$

We define the error of a prediction $\hat{\mathbf{y}}$ as $\eta := \inf_{\text{optimal dual } \mathbf{y}^*} \|\hat{\mathbf{y}} - \mathbf{y}^*\|_1$. In the next subsections, we sketch the ideas to obtain the *usefulness* and *stability* parts of Theorem 1.1. Proofs omitted from this section are found in Appendix B.1.

## 2.1. Algorithm with Dual Predictions

In this section, we show that dual laminar predictions are useful, in the sense that they can improve the performance of online algorithms when they are sufficiently accurate. We present Algorithm 1 as a learning-augmented algorithm for laminar set cover using dual predictions. Algorithm 1 requires an $\mathcal{R}$-competitive classical online algorithm $A$ (without predictions) as part of its input. Our algorithm and its performance guarantee is also parameterized by a constant $0 < \alpha < 1$. We define the notion of $\alpha$-saturation to capture how close a dual constraint is to being tight.

**Definition 2.2** ($\alpha$-saturated). For any $\alpha \in (0, 1)$, we say $S$ is $\alpha$-saturated by a solution $\mathbf{y}$ if $\sum_{e \in S} y_e \geq \alpha \cdot c_S$.

---

**Algorithm 1** Algorithm with dual predictions for laminar set cover

---

1:  Receive prediction $\hat{\mathbf{y}} \in \mathbb{R}^{\mathcal{U}}$ for an optimal dual solution.
2:  $\mathbf{x}^1 \leftarrow \mathbf{0} \in \mathbb{R}^{\mathcal{S}}$, $\mathbf{x}^2 \leftarrow \mathbf{0} \in \mathbb{R}^{\mathcal{S}}$, $\mathbf{x} \leftarrow \mathbf{0} \in \mathbb{R}^{\mathcal{S}}$
3:  Initialize classical online algorithm $A$, with competitive ratio $\mathcal{R}$.
4:  **for** each $e_t$ not yet covered by $\mathbf{x}$ **do**
5:     **if** $\exists S \in \mathcal{S}$ with $e_t \in S$ being $\alpha$-saturated by $\hat{\mathbf{y}}$ **then**
6:        Let $S$ be the inclusion-wise maximal such set
7:        $\mathbf{x}_S^1 \leftarrow 1$ {Type 1 Purchase}
8:     **else**
9:        Pass $e_t$ into $A$.
10:       Let $\mathbf{x}^2$ be the output of $A$. {Type 2 Purchase}
11:    **end if**
12:    $\mathbf{x} \leftarrow \mathbf{x}^1 + \mathbf{x}^2$
13: **end for**

---

The following theorem implies the usefulness property of Theorem 1.1 by choosing $\alpha = 1/(1 + \epsilon)$.

**Theorem 2.3.** *The cost of Algorithm 1 is bounded by*

$$\mathbb{E}[\text{ALG}] \leq \frac{1}{\alpha}\text{OPT} + \frac{1}{\alpha}\|(\hat{\mathbf{y}} - \mathbf{y}^*)^+\|_1 + \frac{\mathcal{R}}{1 - \alpha}\|(\mathbf{y}^* - \hat{\mathbf{y}})^+\|_1$$

*for any optimal dual solution* $\mathbf{y}^*$.

As labeled in Algorithm 1 on Lines 7 and 10, we consider two types of costs: Type 1 costs and Type 2 costs. Type 1 costs are for purchasing permits which are $\alpha$-saturated by $\hat{\mathbf{y}}$, whereas Type 2 costs equal the cost of $A$. Thus, we break the analysis into two parts, bounding above the Type 1 costs and Type 2 costs separately. In particular, Theorem 2.3 follows from Lemma 2.4 and Lemma 2.5.

**Lemma 2.4.** *The Type 1 costs of Algorithm 1 are at most* $\frac{1}{\alpha}\text{OPT} + \frac{1}{\alpha}\|(\hat{\mathbf{y}} - \mathbf{y}^*)^+\|_1$.

*Sketch of Proof.* The high level idea is that the costs from Type 1 purchases either correspond to the optimal dual solution $\mathbf{y}^*$, or are due to some surplus $\|(\hat{\mathbf{y}} - \mathbf{y}^*)^+\|_1$ which

caused a set to be $\alpha$-saturated. We can thus charge these costs to those corresponding objects, losing a factor of $1/\alpha$ in the process. As $\|\mathbf{y}^*\|_1 \leq \text{OPT}$, we obtain the result. $\square$

**Lemma 2.5.** *The Type 2 costs of Algorithm 1 are at most* $\frac{\mathcal{R}}{1-\alpha}\|(\mathbf{y}^* - \hat{\mathbf{y}})^+\|_1$.

*Sketch of Proof.* By complementary slackness, if a requested element is not part of an $\alpha$-saturated constraint, this implies a large error in $\|(\mathbf{y}^* - \hat{\mathbf{y}})^+\|_1$. Therefore, whenever we need to use the Type 2 purchases, we can charge this cost to corresponding parts of $(\mathbf{y}^* - \hat{\mathbf{y}})^+$, losing a factor of $\mathcal{R}/(1 - \alpha)$ in the process. $\square$

## 2.2. Stability of Optimal Dual Solutions

To prove stability, we present Algorithm 2 as a simple algorithm for calculating an optimal dual solution for an offline instance. It is a greedy algorithm which descends the laminar set hierachy $\mathcal{S}$ in a depth-first manner, and increases all dual variables in the leaves in $X$ uniformly until doing so would cause a dual constraint to be violated. We say a set $S \in \mathcal{S}$ is *saturated* if $\sum_{e:e \in S \cap X} y_e = c_S$. The algorithm concludes when all dual variables $y_e$ with $e \in X$ belong to a saturated set.

---

**Algorithm 2** Optimal Dual Computation

---

1:  Initialize $\mathbf{y} \leftarrow \mathbf{0}$.
2:  Let $X \subseteq \mathcal{U}$ be the set of elements which must be covered, revealed offline.
3:  Let $\mathcal{L}$ be a depth-first search ordering of leaves in $\mathcal{T}$.
4:  **for** $S \in \mathcal{L}$ with $S \cap X$ non-empty **do**
5:     Increase all $y_e$ with $y_e \in S \cap X$ uniformly until an ancestor $A$ of $S$ becomes saturated.
6:  **end for**

---

**Lemma 2.6.** *Algorithm 2 computes an optimal solution to the dual program.*

Let $\mathbf{y}^{\text{ALG}}(X)$ denote the result of Algorithm 2 on problem instance $X$. Our next result shows that $\mathbf{y}^{\text{ALG}}(X)$ is stable under perturbations of $X$.

**Theorem 2.7.** *Algorithm 2 satisfies* $\|\mathbf{y}^{\text{ALG}}(X) - \mathbf{y}^{\text{ALG}}(X')\|_1 \leq 2\beta |X \Delta X'|$, *where* $X \Delta X' := (X - X') \cup (X' - X)$ *denotes symmetric difference, and* $\beta := \max_{e \in X \Delta X'} \min_{S:e \in S} c_S$.

## 3. Metrical Task Systems

In Metrical Task Systems (MTS), an algorithm controls a server initially located at a given state (i.e., point) $s_0$ in a metric space $\mathcal{M} = (M, d)$. At each time $t \in \{1, \ldots, T\}$, a request arrives in the form of a cost function $c_t \colon M \rightarrow \mathbb{R}_{\geq 0} \cup \{\infty\}$. In response, the algorithm moves the server to

a new state $s_t \in M$, paying movement cost $d(s_{t-1}, s_t)$ and service cost $c_t(s_t)$. The goal is to minimize the total cost incurred while serving the requests.

We can formulate MTS as an integer program by introducing variables $f_t(a, b) \in \{0, 1\}$ for $a, b \in M$, $t = 1, \ldots, T$, where $f_t(a, b) = 1$ means that the server is moved from state $a$ to state $b$ at time $t$ (just before serving request $c_t$). By relaxing the constraints $f_t(a, b) \in \{0, 1\}$ to $f_t(a, b) \geq 0$, we obtain the following linear program:

$$\min \quad \sum_{t=1}^{T} \sum_{a,b \in M} f_t(a, b)\big(d(a, b) + c_t(b)\big) \qquad \text{(Obj)}$$

$$\text{s.t.} \quad \sum_{b \in M} f_1(a, b) = \mathbf{1}[a = s_0] \qquad \forall a \in M \quad \text{(C1)}$$

$$\sum_{b \in M} \big(f_{t+1}(a, b) - f_t(b, a)\big) = 0 \quad \forall a \in M, \, t \in [T-1]$$
$$\text{(C2)}$$

$$f_t(a, b) \geq 0 \quad \forall a, b \in M, \, t \in [T] \qquad \text{(C3)}$$

Here, $f_t(a, b)$ represents the amount of server mass moved from $a$ to $b$ at time $t$, and $\sum_a f_t(a, b)$ represents the server mass located at $b$ at time $t$. The constraints (C1) ensure that all the server mass is initially located at $s_0$, while the constraints (C2) ensure the conservation of server mass: mass coming into state $a$ at time $t$ must equal the mass leaving state $a$ at time $t + 1$.

By introducing variables $w_0(a)$ and $w_t(a)$, corresponding to the constraints (C1) and (C2), respectively, we obtain the following dual program:

$$\max \quad w_0(s_0)$$

$$\text{s.t.} \quad w_{t-1}(a) \leq w_t(b) + c_t(b) + d(a, b),$$
$$\forall a, b \in M \quad \forall t \in [T-1],$$

$$w_{T-1}(a) \leq c_T(b) + d(a, b), \quad \forall a, b \in M.$$

Observe that the optimal solution of the dual program can be computed in decreasing order of $t$, by setting each variable to the highest possible value which does not violate any constraint. By defining $w_T(a) = 0$ for all $a \in M$, we obtain the following recurrence for the optimal solution:

$$w_{t-1}(a) = \min_{b \in M}\{d(a, b) + c_t(b) + w_t(b)\},$$
$$\forall t \in \{1, \ldots, T\}, \forall a \in M.$$

Notably, $w_t(a)$ precisely captures the minimum possible cost to serve requests at times $t + 1, \ldots, T$, if the server is at state $a$ after serving the request at time $t$. In this sense, $w_t$ can be viewed as a time-reversed analogue of a

work function, a central object in the study of online algorithms (Chrobak & Larmore, 1991). While a work function summarizes the minimum cost of serving the requests revealed so far and can therefore be computed online, the optimal dual variables in our formulation depend on future requests and hence cannot be computed online.

### 3.1. Algorithm with Dual Predictions

Our algorithm using predictions of the dual variables $w_t(s)$ is very simple, and resembles the well-known $A^*$ search algorithm. Algorithm 3 attempts to minimize the actual cost paid in the next move, plus the estimated future cost.

---

**Algorithm 3** Algorithm with dual predictions for MTS

1: **for** $t = 1, \ldots, T$ **do**
2:     Receive cost function $c_t$ and prediction $\widehat{w}_t$
3:     Choose $s_t \in \operatorname{argmin}_{s \in M} d(s_{t-1}, s) + c_t(s) + \widehat{w}_t(s)$
4: **end for**

---

Interestingly, if instead of $\widehat{w}_t$ we used the work function corresponding to time $t$ (which corresponds to dual variables of a *different* LP formulation), we would recover precisely the Work Function Algorithm, which is the optimal deterministic online algorithm for MTS (Chrobak & Larmore, 1998).

Before analyzing the performance of our algorithm, we first describe how we measure the prediction error. We assume that the prediction at the last step is always the zero function (i.e., $\widehat{w}_T = 0$), as the algorithm knows that the online instance ends after the request $c_T$.

Let $B^c$ be the Bellman operator with cost function $c$:

$$(B^c(w))(s) = \min_{s' \in M}\{w(s') + d(s, s') + c(s')\},$$

and let $\|\cdot\|_{\mathrm{sp}}$ denote the span seminorm

$$\|v\|_{\mathrm{sp}} := \max(v) - \min(v),$$

where $\max(v) = \max_i v_i$ and $\min(v) = \min_i v_i$.

For an instance with cost functions $c_1, \ldots, c_T$, we define the error of the prediction $\widehat{w}$ by

$$\eta_t = \|B^{c_t}(\widehat{w}_t) - \widehat{w}_{t-1}\|_{\mathrm{sp}} \quad \text{and} \quad \eta = \sum_{t=1}^{T} \eta_t.$$

Intuitively, $\eta_t$ quantifies the discrepancy between $\widehat{w}_{t-1}$, our estimate of the future cost prior to observing the request $c_t$, and $B^{c_t}(\widehat{w}_t)$, the corresponding estimate after observing $c_t$. We use the more forgiving norm $\|\cdot\|_{\mathrm{sp}}$ rather than $\|\cdot\|_\infty$, as each $\widehat{w}_t$ only needs to be defined up to an additive constant, which has no impact on the behavior of the algorithm. Note that the optimal dual solution $w$ satisfies $w_{t-1} = B^{c_t}(w_t)$ by definition, and therefore yields $\eta = 0$.

**Theorem 3.1.** *Algorithm 3 is $\left(1 + \frac{\eta}{\text{OPT}}\right)$-competitive.*

*Proof.* By choice of $s_t$,

$$d(s_{t-1}, s_t) + c_t(s_t) + \widehat{w}_t(s_t) = (B^{c_t}(\widehat{w}_t))(s_{t-1}).$$

Rearranging and summing over $t$, we obtain

$$
\begin{aligned}
\text{ALG} &= \sum_{t=1}^{T} d(s_{t-1}, s_t) + c_t(s_t) \\
&= \sum_{t=1}^{T} \left((B^{c_t}(\widehat{w}_t))(s_{t-1}) - \widehat{w}_t(s_t)\right) \\
&= \widehat{w}_0(s_0) - \widehat{w}_T(s_T) + \sum_{t=1}^{T} (B^{c_t}(\widehat{w}_t) - \widehat{w}_{t-1})(s_{t-1}) \\
&\leq \widehat{w}_0(s_0) - \widehat{w}_T(s_T) + \sum_{t=1}^{T} \max\left(B^{c_t}(\widehat{w}_t) - \widehat{w}_{t-1}\right).
\end{aligned}
\tag{1}
$$

On the other hand, if the offline optimum solution is $s_0^* = s_0, s_1^*, \ldots, s_T^*$, we have

$$d(s_{t-1}^*, s_t^*) + c_t(s_t^*) + \widehat{w}_t(s_t^*) \geq (B^{c_t}(\widehat{w}_t))(s_{t-1}^*),$$

and by similar calculations we obtain

$$
\text{OPT} \geq \widehat{w}_0(s_0^*) - \widehat{w}_T(s_T^*) + \sum_{t=1}^{T} \min\left(B^{c_t}(\widehat{w}_t) - \widehat{w}_{t-1}\right).
\tag{2}
$$

Since $s_0 = s_0^*$ and $w_T(s_T) = w_T(s_T^*) = 0$, combining (1) and (2) gives

$$
\text{ALG} \leq \text{OPT} + \sum_{t=1}^{T} \|B^{c_t}(\widehat{w}_t) - \widehat{w}_{t-1}\|_{\text{sp}} = \text{OPT} + \eta. \quad \square
$$

### 3.2. Stability of Dual Predictions

The following theorem shows that predictions optimized for nearby instances have small error.

**Theorem 3.2.** *Suppose that, on an instance with cost functions $c_1, \ldots, c_T$, we use predictions $\widehat{w}_1, \ldots, \widehat{w}_T$, which represent the optimal dual solution for an instance with cost functions $\widehat{c}_1, \ldots, \widehat{c}_T$. Then, $\eta \leq \sum_{t=1}^{T} \|\widehat{c}_t - c_t\|_{\text{sp}}$.*

*Proof.* Fix $t \in \{1, \ldots, T\}$. We know that $\widehat{w}_{t-1} = B^{\widehat{c}_t}(\widehat{w}_t)$, and therefore

$$
\begin{aligned}
\eta_t &= \|B^{c_t}(\widehat{w}_t) - \widehat{w}_{t-1}\|_{\text{sp}} \\
&= \|B^{c_t}(\widehat{w}_t) - B^{\widehat{c}_t}(\widehat{w}_t)\|_{\text{sp}}.
\end{aligned}
$$

We can now bound $\eta_t$ by $\|\widehat{c}_t - c_t\|_{\text{sp}}$. Fix $s \in M$ and let $r \in M$ such that

$$(B^{\widehat{c}_t}(\widehat{w}_t))(s) = \widehat{w}_t(r) + d(s, r) + \widehat{c}_t(r).$$

Then

$$
\begin{aligned}
(B^{c_t}(\widehat{w}_t))(s) &\leq \widehat{w}_t(r) + d(s, r) + c_t(r) \\
&= (B^{\widehat{c}_t}(\widehat{w}_t))(s) + c_t(r) - \widehat{c}_t(r) \\
&\leq (B^{\widehat{c}_t}(\widehat{w}_t))(s) + \max(c_t - \widehat{c}_t).
\end{aligned}
\tag{3}
$$

One can similarly show that

$$(B^{\widehat{c}_t}(\widehat{w}_t))(s) \leq (B^{c_t}(\widehat{w}_t))(s) - \min(c_t - \widehat{c}_t). \tag{4}$$

Since (3) and (4) hold for all $s \in M$, we indeed have $\eta_t \leq \|c_t - \widehat{c}_t\|_{\text{sp}}$. $\square$

## 4. Experiments

### 4.1. Laminar Set Cover (Parking Permit Problem)

We conduct an empirical evaluation of our laminar set cover algorithm by instantiating it on the Parking Permit Problem (PPP). Canonically in PPP, we consider an employee who must drive to work on rainy days, and otherwise they walk. When they drive to work, they must possess a valid parking permit to use the parking space. The dilemma is that the parking space has $K$ different types of permits, where each permit type $k = 1, \ldots, K$ lasts some fixed duration $D_k$ of consecutive days and costs some fixed amount $C_k$. The employee must decide whether it is more cost effective to purchase more expensive permits of longer durations, which may be cheaper on a per-rainy-day basis. The problem is a special case of online set-cover, where the days are elements of the universe, and the sets correspond to the days covered by each possible permit. Meyerson (2005) showed that the PPP can be reduced to Laminar Set Cover, while only losing a constant multiplicative factor in the competitive ratio.

While PPP is a general rent-or-buy problem, in our experiments we emulate the above scenario. We use 153 years of weather data measured between 1869 and 2021 at Central Park, New York City, acquired from the Global Historical Climatology Network daily (GHCNd) dataset (NOAA National Centers for Environmental Information, 2023).

We create problem instances consisting of $T = 365$ days, motivated by the seasons of the year. We calculate the optimal dual vector for each instance, and use the point-wise sample mean over training instances as the predictor $\hat{\mathbf{y}}$. We evaluate the performance, as measured by the competitive ratio, using *leave-one-out* cross validation, by leaving out each instance from the dataset and training on the others.

The permit types in our instances have $D_k = 2^k$ and $C_k = (2/f)^k$, where $f$ is a discount factor, representing

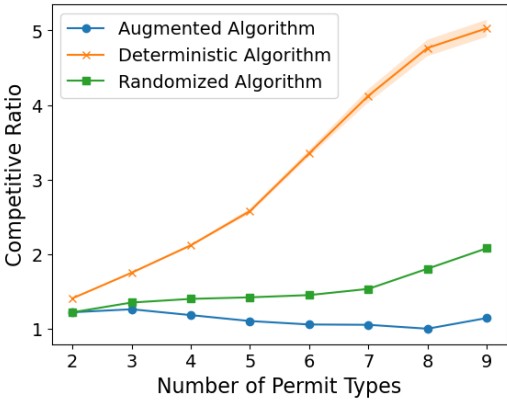

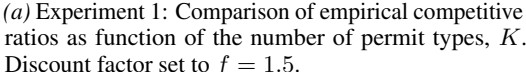

*(a)* Experiment 1: Comparison of empirical competitive ratios as function of the number of permit types, $K$. Discount factor set to $f = 1.5$.

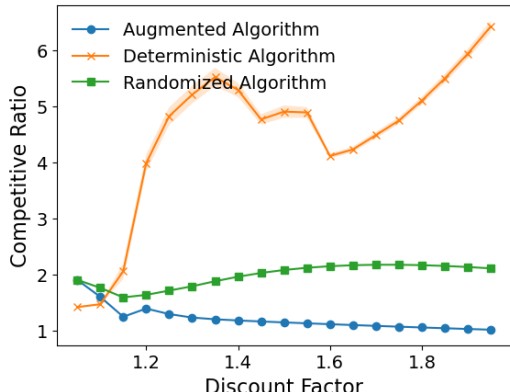

*(b)* Experiment 2: Comparison of empirical competitive ratios obtained as a function of the Discount factor, $f$. Number of permits $K$ set to $K = 9$.

*Figure 1.* Experiments on the Parking Permit Problem. Shades denote approximate 95% confidence intervals.

the factor by which costs of subsequent permits decrease. We compare the performance of our learning-augmented algorithm (with $\alpha = 0.5$) relative to state-of-the-art $O(K)$-competitive deterministic and $O(\log K)$-competitive randomized classical online algorithms for PPP. We use the randomized algorithm as the input $\mathcal{R}$-competitive algorithm in our learning-augmented procedure.

The experimental results are displayed in Figures 1a and 1b. In Figure 1a, we show how the empirical competitive ratio depends on $K$ for the different algorithms. We observe that the learning-augmented algorithm consistently has the best performance, and is near-optimal. The gap between the algorithms is more substantial as $K$ increases, where at $K = 9$, the randomized algorithm has a competitive ratio $1.8$ times greater than the learning-augmented algorithm, and the deterministic algorithm is $4.4$ times greater. Figure 1b explores the relationship between the discount factor and the algorithm's performance. For smaller discount factors of $f = 1.1$ or less, the deterministic algorithm performs better, likely because there is less penalty for conservatively choosing the cheaper permits. As $f$ grows, the gap between each algorithm grows, but this is not strictly monotonic. The conclusion from these experiments is that the learning-augmented algorithm is consistently useful on real-world data, particularly when the number of permits or the discount factors are large.

### 4.2. Metrical Task Systems (k-Server)

We evaluate the practicality of dual-based predictions for a particularly important special case of MTS, namely the k-server problem (Manasse et al., 1988). In this problem, there are $k$ servers in a metric space, and requests arrive online at points in this metric space. Each request $r_t$ must be serviced by moving one of the servers to the requested location, and

the goal is to minimize the total distance traveled by the servers. Note that $k$-server reduces to MTS on the metric space of server configurations, and where the cost function $c_t$ is 0 at configurations containing $r_t$ and $\infty$ elsewhere.

We conduct our experiments using a real-world dataset from a popular bike-sharing platform (cit, 2025), which was previously used for benchmarks on a particular case of the $k$-server problem, namely caching (Lykouris & Vassilvit-skii, 2021; Antoniadis et al., 2023c). We focus on trips originating in Manhattan and map each trip to a point in a metric space of size 10. Specifically, we partition Manhattan into 10 equally-sized intervals of latitude, which we associate to 10 points equally-spaced on a line, and then map each trip to the point corresponding to the latitude of its starting station. Using this mapping, we construct a $k$-server instance for each day.

We perform two experiments on this dataset. In the first one, we issue a separate request for each individual trip. In the second experiment, we issue one request per minute of the day, corresponding to the point associated with the largest number of trips in that minute. In both experiments, we use data from 2023 and 2024 for training (731 instances in total) and data from 2025 for testing (365 instances).

To learn predictions, we first compute the optimal dual variables for each training instance. We then partition time into blocks of 15 minutes and, for each instance and each block, compute the average of the optimal dual variables over that block. Next, we average these values across all training instances, yielding a single prediction per 15-minute block. During testing, we provide these learned average dual variables as predictions, updating them every 15 minutes. For example, all requests arriving between 1:00 and 1:15 are assigned the prediction obtained by averaging the dual vari-

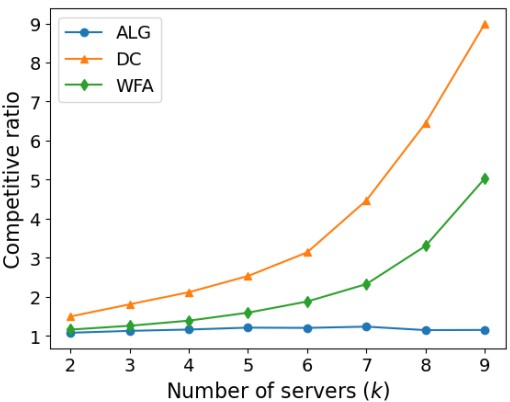

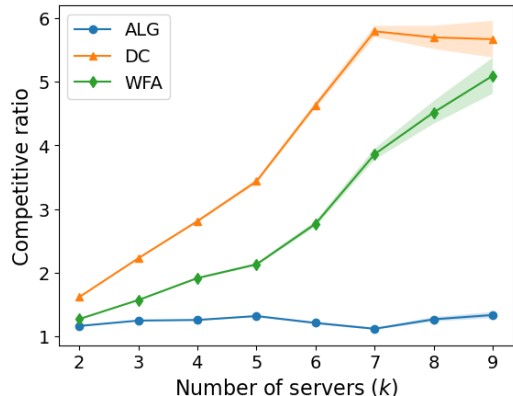

*(a)* Experiment 1: A separate request for each trip.

*(b)* Experiment 2: One request per minute, at the point associated with the largest number of trips.

*Figure 2.* The average competitive ratio of each algorithm, where the number of points in the metric space is fixed to 10, and the number of servers varies from $k = 2$ to $k = 9$. Shades denote approximate 95% confidence intervals.

ables corresponding to the 1:00–1:15 time block across all training instances. This choice of prediction reflects the idea that request patterns remain relatively stable within short time intervals. Moreover, aggregating predictions over 15-minute blocks yields sparser and more interpretable predictions, while also improving statistical robustness, since each learned prediction is based on multiple optimal dual solutions, both across days and within a single day.

We compare the performance of our algorithm with two classical online algorithms: the Work Function Algorithm (WFA) and the Double Coverage (DC) algorithm. Both of them are known to achieve the optimal competitive ratio $k$ among deterministic online algorithms for the line metric space (Chrobak et al., 1990; Bartal & Koutsoupias, 2004).

The experimental results are shown in Figure 2. We observe that our algorithm consistently outperforms the classical online algorithms. Moreover, while our algorithm is nearly optimal for any number of servers $k$, the performance of WFA and DC degrades as function of $k$, in line with their theoretical $k$-competitiveness guarantees.

## 5. Conclusion

In this work, we demonstrated that for a rich assortment of problems, dual predictions possess several advantages over other typical prediction types. We emphasize that the predictions considered by learning-augmented algorithms should not be so strong and difficult to learn that it merely outsources the difficult task to upstream learning algorithms. Our learnability and stability results for our proposed predictions take into account the practical limitations of modern machine-learned predictors, making them more practical than many previous models.

Future work could explore whether similar results can be obtained for problems that do not fall into either MTS or laminar set cover. For general set cover, at least when using its standard LP formulation, there are worst-case instances that remain hard even when an optimal dual solution is known upfront (cf. Appendix A.3). But it is an interesting question whether a different way of modeling set cover might lead to more useful duals (similarly to the situation for MTS, where our results rely on our particular LP formulation, whereas a very similar LP formulation for MTS would have been unsuitable, as discussed in Section 3.1).

## Acknowledgements

We thank the anonymous reviewers for their valuable comments.

Funded by the European Union (ERC, CCOO, 101165139). Views and opinions expressed are however those of the author(s) only and do not necessarily reflect those of the European Union or the European Research Council. Neither the European Union nor the granting authority can be held responsible for them.

## Impact Statement

This paper presents work whose goal is to advance the field of learning-augmented algorithms. There are many potential societal consequences of our work, none which we feel must be specifically highlighted here.

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

# A. Further Theoretical Results

In this section, we present some further results, which serves to further contextualize our research. In Appendix A.1, we show that primal predictions do not have the same stability guarantees as dual predictions. In Appendix A.2, we show that a popular prediction model for caching is unstable. Finally, in Appendix A.3, we show that for general (non-laminar) set cover, even with perfect optimal dual predictions, a learning-augmented algorithm cannot do better than the classical online algorithm in some cases, at least for the natural LP formulation.

## A.1. Instability of Primal Predictions

Firstly, we showcase that optimal primal predictions often do not enjoy the same stability as optimal dual predictions. We show that for optimal primal predictions, there is no analogous stability theorem to Theorem 2.7.

**Lemma A.1.** *There exist problem instances $X$ and $X'$ for laminar set cover, such that for any primal optima $\mathbf{x}$ and $\mathbf{x}'$, $\|\mathbf{x} - \mathbf{x}'\|_1 / |X \Delta X'|$ can be made arbitrarily large, even when $\beta := \max_{e \in \mathcal{U}} \min_{S : e \in S} c_S = 1$.*

*Proof.* We use the notation as defined in the main text. Firstly, let $N$ be an arbitrarily large integer. We consider an $(N+1)$-element universe $\mathcal{U} = \{0, 1, \dots, N\}$, and a collection of subsets $\mathcal{S} = \{\mathcal{U}, S_0, \dots, S_N\}$, where $S_i = \{i\}$ for $0 \le i \le N$. Then, define $c(\mathcal{U}) = N + \frac{1}{2}$, whereas $c(S_i) = 1$. Observe that for all $N$, $\beta$ is fixed at 1.

Now, consider $X = \mathcal{U}$ and $X' = \mathcal{U} - \{0\}$. Then $|X \Delta X'| = 1$. Observe that for $X$, the unique optimal solution is to buy $\mathcal{U}$, and on the other hand, the only optimal solution for $X'$ is $= \{S_1, \dots, S_N\}$. Therefore, the primal solution vectors $\mathbf{x}$ and $\mathbf{x}'$ differ in $N + 1$ positions, meaning $\|\mathbf{x} - \mathbf{x}'\|_1 = N + 1$. Thus, $\|\mathbf{x} - \mathbf{x}'\|_1 / |X \Delta X'| = N + 1$, and can therefore be made arbitrarily large. □

Next, we show that primal predictions are also unstable for $k$-server, which is a special case of MTS. In this problem, $k$ servers must be moved in a metric space $\mathcal{M} = (M, d)$ to serve incoming requests. At each time $t = 1, \dots, T$, a request consisting of a point $r_t \in M$ arrives, and one of the servers must be moved to $r_t$. The cost incurred is equal to the total distance traveled by the servers. We can reduce $k$-server to MTS on the metric space that contains all possible configurations, and the service costs are 0 at configurations containing $r_t$ and $\infty$ elsewhere.

**Lemma A.2.** *There exist problem instances $(r_t)_{t \ge 0}$ and $(r'_t)_{t \ge 0}$ for $k$-server such that $(r_t)$ and $(r'_t)$ differ in a single request, and the difference $\|f - f'\|_1$ in the corresponding optimal primal solutions can be made arbitrarily large.*

*Proof.* Consider the 3-server problem on the line with all servers starting at 0, and the request sequence repeats $-2, -1, 2, 1$ many times and ends with a final request at $-2$. Then the unique optimal solution keeps two servers stationary at $-2$ and $-1$ and the third server goes back and forth between 1 and 2. Changing the final request to 2 completely changes/mirrors the unique optimal solution, with two servers stationary at 1 and 2 and the third going back and forth between $-2$ and $-1$. □

## A.2. Instability of Event Predictions for Caching

Perhaps the most well-known line of work within learning-augmented algorithms is Lykouris and Vassilvitskii's model for paging (special case of MTS) with next-request time prediction. In this model (Lykouris & Vassilvitskii, 2021), each request to a page $p$ comes with a prediction of the next time that page $p$ is requested again. The best learning-augmented algorithm for this setting by Wei (2020) works as follows: on a cache miss, evict the page with the furthest-in-future *predicted* next-request time, but default to a standard online algorithm ignoring predictions if it has smaller overall cost.

We show that these predictions are also unstable in a very strong sense.

**Lemma A.3.** *Replacing just a* single *request in an otherwise perfectly predicted sequence can lead not only to unbounded prediction error, but to the performance of the learning-augmented algorithm by Wei (2020) completely deteriorating to a competitive ratio of $\Omega(\log k)$, offering no improvement over an online algorithm without predictions.*

*Proof.* Consider the following request sequence:

- Stage 1: Request $1, 2, \dots, k, 2$.

- Stage 2: Repeatedly request $2, 3, \dots, k+1$ many times.

- Stage 3: An adversarial sequence on the pages $2, 3, \ldots, k + 2$ such that the online algorithm without predictions attains its worst-case competitive ratio ($\Omega(\log k)$ if randomized, $\Omega(k)$ if deterministic).

Now imagine the *anticipated* request sequence is the same except the last request of Stage 1 is 1 instead of 2, and next-request time predictions are sampled according to this anticipated sequence.

The first request, at page 1, is accompanied by a prediction that page 1 will be requested again at time $k + 1$. Thus, after the first $k$ requests fill the initially empty cache, the evict-furthest-predicted policy would refuse to ever evict page 1 as it anticipates a next request to 1 earlier than any other request (although in the true request sequence page 1 is never requested again). But this would lead to cost tending to infinity during Stage 2, whereas the online algorithm without predictions would serve Stage 2 for constant cost by keeping pages $2, 3, \ldots, k + 1$ in cache. This triggers the defaulting behavior, whereby predictions are ignored and the overall algorithm offers no improvement over the online algorithm without predictions. □

### A.3. Difficulty of Using Dual Predictions for General Set Cover

In this section, we show that for the standard LP formulation of fractional set cover (i.e the same as the laminar set cover formulation in the main text), predictions of the optimal dual are not useful, in the sense of the following result.

**Lemma A.4.** *A learning-augmented algorithm for online fractional set cover, with perfect predictions of the optimal dual, has competitive ratio of at least $H_m$, where $H_k$ is the kth Harmonic number and $m$ is the number of sets, $|\mathcal{S}|$.*

*Proof.* We first present a proof of how a classical $H_m$ lower-bound on fractional set cover can be found (cf. Alon et al. (2009)), and then show that perfect dual predictions are unhelpful at improving this.

Fix a deterministic online algorithm $\mathcal{A}$. We exhibit an adaptive adversarial instance with $m$ sets $S_1, \ldots, S_m$ with $c(S_i) = 1$, on which $\mathcal{A}$ incurs cost at least $H_m$ while OPT $= 1$. For our construction, we require that for any non-empty subset of the sets $S_1, \ldots, S_m$, there exists an element which appears precisely in those subsets.

*Construction.* The adversary maintains a nested sequence of *alive* index sets

$$A_0 \supsetneq A_1 \supsetneq \cdots \supsetneq A_{m-1},$$

with $A_0 = [m]$ and $|A_k| = m - k$. Let $x^{(k)} \in \mathbb{R}^m_{\geq 0}$ denote the algorithm's variable vector immediately after phase $k$, with $x^{(0)} = \mathbf{0}$. The vectors are nondecreasing: $x^{(k)} \geq x^{(k-1)}$ componentwise.

For phases $k = 1, 2, \ldots, m - 1$:

1. The adversary selects

$$i_k \in \arg \max_{i \in A_{k-1}} x_i^{(k-1)},$$

   breaking ties arbitrarily, and sets $A_k := A_{k-1} \setminus \{i_k\}$.

2. The adversary presents element $e_k$ with

$$\{i \in [m] : e_k \in S_i\} = A_k.$$

   The algorithm responds, producing $x^{(k)} \geq x^{(k-1)}$ satisfying

$$\sum_{i \in A_k} x_i^{(k)} \geq 1. \tag{$*_k$}$$

Finally, let $i_m$ denote the unique element of $A_{m-1}$. The adversary presents one last element $e_m$ with $\{i : e_m \in S_i\} = \{i_m\}$, forcing $x_{i_m}^{(m)} \geq 1$.

*Bounding* OPT. By construction, $i_m \in A_k$ for every $k \in \{0, 1, \ldots, m - 1\}$, hence $e_k \in S_{i_m}$ for all $k \in [m]$. The offline assignment $x_{i_m}^* = 1$ and $x_i^* = 0$ for $i \neq i_m$ is therefore feasible and hence OPT $= 1$.

*Bounding* ALG. We may assume without loss of generality that $\mathcal{A}$ is *minimal* in the sense that at the end of each phase $k \in [m - 1]$,

$$\sum_{i \in A_k} x_i^{(k)} = 1,$$

i.e., the algorithm raises variables only as much as needed to satisfy $(*_k)$; any algorithm that overshoots only incurs greater cost, so a lower bound under this assumption implies the same lower bound in general.

We claim that for every $k \in \{1, \ldots, m-1\}$,

$$\sum_{i \in [m]} \left( x_i^{(k)} - x_i^{(k-1)} \right) \geq \frac{1}{m-k+1}. \tag{$\dagger_k$}$$

*Case $k = 1$.* Here $x^{(0)} = \mathbf{0}$ and $(*_1)$ gives $\sum_{i \in A_1} x_i^{(1)} \geq 1$, so $\sum_i x_i^{(1)} \geq 1 \geq \frac{1}{m}$.

*Case $k \geq 2$.* By the minimality assumption applied to phase $k-1$,

$$\sum_{i \in A_{k-1}} x_i^{(k-1)} = 1. \tag{1}$$

Since $i_k$ maximizes $x_i^{(k-1)}$ over $i \in A_{k-1}$ and $|A_{k-1}| = m-k+1$,

$$x_{i_k}^{(k-1)} \geq \frac{1}{m-k+1} \sum_{i \in A_{k-1}} x_i^{(k-1)} = \frac{1}{m-k+1}. \tag{2}$$

Since $A_k = A_{k-1} \setminus \{i_k\}$, combining (1) and (2),

$$\sum_{i \in A_k} x_i^{(k-1)} = \sum_{i \in A_{k-1}} x_i^{(k-1)} - x_{i_k}^{(k-1)} \leq 1 - \frac{1}{m-k+1}. \tag{3}$$

Constraint $(*_k)$ requires $\sum_{i \in A_k} x_i^{(k)} \geq 1$, so

$$\sum_{i \in A_k} \left( x_i^{(k)} - x_i^{(k-1)} \right) \geq 1 - \sum_{i \in A_k} x_i^{(k-1)} \geq \frac{1}{m-k+1},$$

and since $x^{(k)} \geq x^{(k-1)}$ componentwise, this yields $(\dagger_k)$.

The final element $e_m$ forces $x_{i_m}^{(m)} \geq 1$. By minimality, $x_{i_m}^{(m-1)}$ contributes to $\sum_{i \in A_{m-1}} x_i^{(m-1)} = 1$, but $A_{m-1} = \{i_m\}$, so $x_{i_m}^{(m-1)} = 1$ already and the final element adds no further cost. (If instead one prefers to terminate at phase $m-1$, the bound $H_m - 1 + 1 = H_m$ still emerges from the analysis below; the singleton element is for definiteness.)

*Summing.* Since the unit costs give $\mathrm{ALG} = \sum_i x_i^{(m)}$ and the increments telescope,

$$\mathrm{ALG} = \sum_{k=1}^{m} \sum_{i \in [m]} \left( x_i^{(k)} - x_i^{(k-1)} \right) \geq \sum_{k=1}^{m-1} \frac{1}{m-k+1} + \left( x_{i_m}^{(m)} - x_{i_m}^{(m-1)} \right).$$

The first sum, reindexed by $j = m-k+1$, equals $\sum_{j=2}^{m} \frac{1}{j} = H_m - 1$. Combined with the final increment, which (in the case $x_{i_m}^{(m-1)} < 1$) contributes the missing 1, we obtain

$$\mathrm{ALG} \geq H_m.$$

Combining with $\mathrm{OPT} \leq 1$,

$$\frac{\mathrm{ALG}}{\mathrm{OPT}} \geq H_m.$$

Now, consider the prediction $\hat{\mathbf{y}}$ which is zero, except for being 1 at the index $e_1$. Observe that this dual solution is feasible, and it is optimal as its objective equals 1, which matches the primal OPT described. But, observe that this prediction cannot be used to improve the performance of an algorithm on online fractional set cover on the described adversarial example, because the prediction $\mathbf{y}_{e_1}$ does not reveal any information (since $e_1$ is covered by all sets $S_i$), and the prediction $\hat{\mathbf{y}}$ is constant at zero everywhere else. $\qquad\square$

# B. Omitted Laminar Set Cover Results

## B.1. Proofs Omitted from Main Text

**Lemma 2.4.** *The Type 1 costs of Algorithm 1 are at most $\frac{1}{\alpha}\text{OPT} + \frac{1}{\alpha}\|(\hat{\mathbf{y}} - \mathbf{y}^*)^+\|_1$.*

*Proof.* The main part of the proof is proving that

$$\text{Type 1 costs} \leq \frac{1}{\alpha}\|\hat{\mathbf{y}}\|_1 \tag{5}$$

We show this inequality using a charging strategy. Recall that for a Type 1 purchase for element $e \in X$, we include the set inclusion-wise maximal set $S$ which is $\alpha$-saturated by $\hat{\mathbf{y}}$, i.e $\sum_{f \in S} \hat{y}_f \geq \alpha \cdot c(S)$, equivalently $c(S) \leq \sum_{f \in S} \frac{\hat{y}_f}{\alpha}$. Hence, we can charge the Type 1 costs to $\frac{\hat{\mathbf{y}}}{\alpha}$. Observe that the laminar assumption makes it impossible for any double charges to occur. To see this, observe that if $S$ and $S'$ were two intervals which were both charged to, it is impossible that one is contained in the other, because Algorithm 1 purchases the inclusion-wise maximal $\alpha$-saturated set. However, if neither is contained in one another, then by the laminar assumption, they must be disjoint. Inequality (5) therefore follows.

Then, we can decompose $\|\hat{\mathbf{y}}\|_1$ using the simple inequality

$$\|\hat{\mathbf{y}}\|_1 \leq \|\mathbf{y}^*\|_1 + \|(\hat{\mathbf{y}} - \mathbf{y}^*)^+\|_1. \tag{6}$$

Finally, observe that $\|\mathbf{y}^*\|_1 = \sum_{e \in \mathcal{U}} y_e^* = \sum_{e \in X}^T y_e^*$, which is the dual objective value. By the strong-duality theorem, as $\mathbf{y}^*$ is an optimal solution to the dual, $\|\mathbf{y}^*\|_1$ equals the objective of the optimal solution to the primal program. Thus, $\|\mathbf{y}^*\|_1 = \text{OPT}$. The string of inequalities therefore implies Lemma 2.4:

$$
\begin{aligned}
\text{Type 1 costs} &\leq \frac{1}{\alpha}\|\hat{\mathbf{y}}\|_1 && \text{by Ineq. (5)} \\
&\leq \frac{1}{\alpha}\left(\|\mathbf{y}^*\|_1 + \|(\hat{\mathbf{y}} - \mathbf{y}^*)^+\|_1\right) && \text{by Ineq. (6)} \\
&= \frac{1}{\alpha}\text{OPT} + \frac{1}{\alpha}\|(\hat{\mathbf{y}} - \mathbf{y}^*)^+\|_1.
\end{aligned}
\tag{7}
$$

$\square$

**Lemma 2.5.** *The Type 2 costs of Algorithm 1 are at most $\frac{\mathcal{R}}{1-\alpha}\|(\mathbf{y}^* - \hat{\mathbf{y}})^+\|_1$.*

*Proof.* Firstly, fix an optimal dual solution $\mathbf{y}^*$. Then, let $\mathbf{x}^*$ be an optimal integral solution to the primal problem. Assume without loss of generality that each element $e \in X$ is covered by exactly one set for the cover $\mathbf{x}^*$. Let $X' \subseteq X$ be the collection of requested elements which do not belong to any $\alpha$-saturated set, and are therefore associated with Type 2 costs. Define the set $\mathbf{x}_2^* := \{S \in \mathbf{x}^* : S \cap X' \neq \varnothing\}$.

Now, by complementary slackness conditions, if $\mathbf{x}_S^* = 1$ then $\sum_{e \in S} y_e^* = c(S)$. Therefore, for all $S \in \mathbf{x}_2^*$ we have $\sum_{e \in S} y_s^* = c(S)$. On the other hand, for all $S \in \mathbf{x}_2^*$, the fact that $S \cap X' \neq \varnothing$ for some $e \in S$ implies that there was no $\alpha$-saturated set containing $e$, meaning $\sum_{e \in S} \hat{y}_e < \alpha \cdot c(S)$. Thus, for all $S \in \mathbf{x}_2^*$, we have $\sum_{e \in S} y_e^* - \hat{y}_e \geq (1 - \alpha)c(S)$. This yields

$$
\begin{aligned}
\sum_{S \in \mathbf{x}_2^*} c(S) &\leq \frac{1}{1-\alpha} \sum_{S \in \mathbf{x}_2^*} \sum_{e \in S} y_e^* - \hat{y}_e \\
&\leq \frac{1}{1-\alpha} \sum_{S \in \mathbf{x}_2^*} \sum_{e \in S} (y_e^* - \hat{y}_e)^+ \\
&\leq \frac{1}{1-\alpha} \sum_{e \in \mathcal{U}} (y_e^* - \hat{y}_e)^+ \\
&= \frac{1}{1-\alpha} \|(\mathbf{y}^* - \hat{\mathbf{y}})^+\|_1,
\end{aligned}
\tag{8}
$$

where the third inequality holds as we assumed that for each $e$, $\mathbf{x}^*$ covers $e$ by at most one set. Now, recall that $A$ is an online algorithm which receives some subset of $X'$ as an online sequence, and call the indicator function of this subset $\boldsymbol{\sigma}$. Define

$\text{OPT}_{\sigma}$ as the cost of an offline optimal running on $\boldsymbol{\sigma}$. Observe that $\text{OPT}_{\sigma} \leq \sum_{S \in \mathbf{x}_2^*} c(S)$ as all $e_t$ in $\boldsymbol{\sigma}$ will be covered by $\mathbf{x}_2^*$, making it a feasible solution. As $A$ is run on the instance $\boldsymbol{\sigma}$ and is $\mathcal{R}$-competitive, we know $\mathbb{E}[\text{cost}(A)] \leq \mathcal{R} \cdot \text{OPT}_{\sigma}$. Chaining these inequalities together, we find

$$
\begin{aligned}
\text{Type 2 costs} &= \mathbb{E}[\text{cost}(A(\boldsymbol{\sigma}))] \\
&\leq \mathcal{R} \cdot \text{OPT}_{\sigma} \\
&\leq \mathcal{R} \cdot \sum_{S \in \mathbf{x}_2^*} c(S) \\
&\leq \frac{\mathcal{R}}{1-\alpha} \|(\mathbf{y}^* - \hat{\mathbf{y}})^+\|_1. \quad \square
\end{aligned}
\tag{9}
$$

**Lemma 2.6.** *Algorithm 2 computes an optimal solution to the dual program.*

*Proof.* For the purposes of this proof, let ALG denote Algorithm 2. We show for any $S \in \mathcal{S}$ that if ALG is run on $\mathcal{S}|_S := \{S' \in \mathcal{S} : S' \subseteq S\}$, then ALG calculates the optimal dual on this subproblem. Let $\text{ALG}(S)$ denote the objective value of running ALG on the instance $\mathcal{S}|_S$. We proceed by induction on the height of $S$ in $\mathcal{T}$, denoted $h(S)$.

For the base case when $h(S) = 0$, we are considering the leaves of $\mathcal{T}$. When ALG is run on $\mathcal{S}|_S$, the algorithm will simply increase the variables $y_e$ with $e \in S \cap X$ until the objective value of $c(S)$ is reached. If $S \cap X = \varnothing$, then the algorithm will stop and the objective will simply be $0$. Either scenario is clearly optimal, concluding the base case.

Consider some $S$, assume that the claim holds for heights less than $h(S)$. Now, if we run ALG on $\mathcal{S}|_S$, then we consider two cases:

**Case 1:** if during the running of the algorithm $S$ becomes saturated, then the objective value $c(S)$ is reached, which is optimal dual to the dual constraint that $\sum_{e \in S} y_e \leq c(S)$.

**Case 2:** suppose $S$ never becomes saturated during the running of ALG on $\mathcal{S}|_S$. Then, the dual constraint for $S$ is never tight, meaning the algorithm is equivalent to running ALG on each of the children of $S$ independently. Let $S_1, ..., S_c$ denote the children of $S$, hence $\text{ALG}(S) = \sum_{i=1}^{c} \text{ALG}(S_i)$. Let $\mathbf{y}$ be any feasible dual solution of $\mathcal{S}|_S$, then

$$
\sum_{e \in S \cap X} y_e = \sum_{i=1}^{c} \sum_{e \in S_i \cap X} y_e \leq \sum_{i=1}^{c} \text{ALG}(S_i) = \text{ALG}(S)
\tag{10}
$$

and therefore $\text{ALG}(S)$ is also optimal for this case. $\square$

Let $\mathbf{y}^{\text{ALG}}(X)$ denote the result of Algorithm 2 on problem instance $X$. Our next result shows that $\mathbf{y}^{\text{ALG}}(X)$ is stable under perturbations of $X$, which is vital for the learnability of the optimal dual prediction.

**Theorem 2.7.** *Algorithm 2 satisfies $\|\mathbf{y}^{\text{ALG}}(X) - \mathbf{y}^{\text{ALG}}(X')\|_1 \leq 2\beta |X \Delta X'|$, where $X \Delta X' := (X - X') \cup (X' - X)$ denotes symmetric difference, and $\beta := \max_{e \in X \Delta X'} \min_{S: e \in S} c_S$.*

Algorithm 2 satisfies $\|\mathbf{y}^{\text{ALG}}(X) - \mathbf{y}^{\text{ALG}}(X')\|_1 \leq 2\alpha |X \Delta X'|$, where $X \Delta X' := (X - X') \cup (X' - X)$ denotes symmetric difference, and $\beta := \max_{e \in X \Delta X'} \min_{S: e \in S} c(S)$.

*Proof.* We first consider the case that $X' = X \cup \{e\}$ for some $e \in \mathcal{U} - X$, and seek to show that $\|\mathbf{y}^{\text{ALG}}(X) - \mathbf{y}^{\text{ALG}}(X')\|_1 \leq 2c(S^e)$, where $S^e$ is the leaf node in $\mathcal{T}$ containing $e$. Let $\mathcal{U} = S_0, S_1, ..., S_d = S^e$ be the path from the root to $S^e$ in $\mathcal{T}$, and let $\delta_i = \sum_{e \in S_i} [y^{\text{ALG}}(X') - y^{\text{ALG}}(X)]_e$ for $1 \leq i \leq d$ be the change in objective value between the two solutions. Observe that $\delta_i \geq 0$, as adding an extra element $e$ cannot decrease the objective value of each subproblem.

We decompose $\|\mathbf{y}^{\text{ALG}}(X) - \mathbf{y}^{\text{ALG}}(X')\|_1 = \sum_{i=0}^{d} s_i$, where we charge cost associated with a change in the $f$th coordinate, $|\mathbf{y}^{\text{ALG}}(X)_f - \mathbf{y}^{\text{ALG}}(X')_f|$, to the deepest $S_i$ on the $\mathcal{U}$-$S^e$ path with $f \in S_i$. The crucial observation is that for $0 \leq i \leq d-1$, we have $s_i \leq \delta_{i+1} - \delta_i$, as this quantity represents the amount of mass that dual variables increased after $y_e$ lose due to the addition of $e$ in $X'$. Additionally, $\delta_d \leq s_d$ is clear - the increase in objective value at $\delta_d$ can only happen alongside a corresponding increasing in $s_d$, as all dual variables corresponding to $S^e$ are increased uniformly. Therefore, we have that $\|\mathbf{y}^{\text{ALG}}(X) - \mathbf{y}^{\text{ALG}}(X')\|_1 = \sum_{i=0}^{d} s_i \leq s_d + \delta_d - \delta_0 \leq 2s_d$.

The final step is to establish the bound $s_d \leq c(S^e)$. Let $n$ be the number of elements in $S^e \cap X$. If $n = 0$, then the only non-zero $\mathbf{y}^{\text{ALG}}(X')_f$ for $f \in S_d$ occurs at $f = e$, as $X' \cap S^e = \{e\}$. Therefore, $s_d \leq c(S^e)$. On the other hand, for $n \geq 1$, the final value of $\mathbf{y}^{\text{ALG}}(X')_e$ is at most $\frac{c(S^e)}{n+1}$, and at worst this amount of dual mass is taken from the elements of $X \cap S^e$. Therefore, $s_d \leq \frac{2c(S^e)}{n+1} \leq c(S^e)$, thereby proving the claim.

The fully general case specified by the Lemma follows by repeatedly applying this claim to all elements $e \in X \Delta X'$ and taking the maximum $c(S^e) = \min_{S:e \in S} c(S)$. $\qquad \square$

## B.2. Learnability

In this section, we show that the optimal dual prediction is efficiently PAC-learnable. We fix $\mathcal{U}, \mathcal{S}$ and $c : \mathcal{S} \rightarrow \mathbb{R}_{\geq 0}$. We let $\boldsymbol{\sigma} \in \{0,1\}^{\mathcal{U}}$ encode $X \subseteq \mathcal{U}$, where $\sigma_e = 1$ indicates that $e \in X$. Now, fix a probability distribution $\mathcal{D}$ over the set of problem instances $\boldsymbol{\sigma}$. The performance of our learning-augmented algorithm, when using prediction $\hat{\mathbf{y}}$ when $\mathbf{y}^*$ is a true optimal, has cost bounded above in terms of $\|\hat{\mathbf{y}} - \mathbf{y}^*\|_1$. Therefore, we define the loss function

$$L(\hat{\mathbf{y}}, \boldsymbol{\sigma}) = \|\hat{\mathbf{y}} - \mathbf{y}^*(\boldsymbol{\sigma})\|_1 \tag{11}$$

and seek to find a prediction $\hat{\mathbf{y}}$ minimizing $\mathbb{E}_{\boldsymbol{\sigma} \sim \mathcal{D}}[L(\hat{\mathbf{y}}, \boldsymbol{\sigma})]$. Let $\hat{\mathbf{y}}^* = \arg\min_{\mathbf{y}} [\mathbb{E}_{\boldsymbol{\sigma} \sim \mathcal{D}} L(\mathbf{y}, \boldsymbol{\sigma})]$.

**Theorem B.1.** *There is a learning algorithm that given $\epsilon, \delta > 0$ returns a dual prediction $\hat{\mathbf{y}}$ such that $\mathbb{E}_{\boldsymbol{\sigma} \sim \mathcal{D}}[L(\hat{\mathbf{y}}, \boldsymbol{\sigma})] \leq \mathbb{E}_{\boldsymbol{\sigma} \sim \mathcal{D}}[L(\hat{\mathbf{y}}^*, \boldsymbol{\sigma})] + \epsilon$ with probability at least $1 - \delta$. The algorithm has sampling complexity $O\left(\left(\frac{\beta}{\epsilon}\right)^2 (n \log n + \log(1/\delta))\right)$ where $\beta := \max_{e \in \mathcal{U}} \min_{S:e \in S} c_S$, $n = |\mathcal{U}|$ and time complexity polynomial in $n, \frac{1}{\epsilon}, \frac{1}{\delta}$.*

*Proof.* For every dual solution $\mathbf{y}$, we let $g_{\mathbf{y}}(\boldsymbol{\sigma}) = L(\mathbf{y}, \boldsymbol{\sigma})$, and define $\mathcal{H} = \{g_{\mathbf{y}} : \mathbf{y} \in \mathbb{R}^T\}$. We use several concepts from statistical learning theory to prove Theorem B.1, namely that to prove learnability it is sufficient to upper-bound the *pseudo-dimension* of $\mathcal{H}$.

**Definition B.2** (Pseudo-dimension). Let $\mathcal{H}$ be a class of functions $h : X \rightarrow \mathbb{R}$ and $S = \{x_1, ..., x_s\} \subseteq X$. Then $S$ is shattered by $\mathcal{H}$ if there exists real numbers $r_1, ..., r_s$ so that for all $S' \subseteq S$ there is a function $h \in \mathcal{H}$ such that $f(x_i) \leq r_i$ iff $x_i \in S'$ for all $i$. The pseudo-dimension is the largest $s$ such that there exists an $S \subseteq X$ with $|S| = s$ which is shattered by $\mathcal{H}$.

The relevance of pseudo-dimension comes from the following convergence result.

**Lemma B.3** ((Anthony & Bartlett, 2009; Morgenstern & Roughgarden, 2015; Pollard, 2012)). *Let $\mathcal{D}$ be a distribution over domain $X$ and $\mathcal{H}$ be a class of functions $h : X \rightarrow [0, H]$ with pseudo-dimension $d_{\mathcal{H}}$. For any $\epsilon, \delta > 0$, if we sample i.i.d $\mathbf{x}_1, ..., \mathbf{x}_s \sim \mathcal{D}$ with $s \geq c_0(\frac{H}{\epsilon})^2(d_{\mathcal{H}} + \ln(1/\delta))$ it holds that*

$$\left| \frac{1}{s} \sum_{i=1}^{s} h(\mathbf{x}_i) - \mathbb{E}_{\mathbf{x} \sim \mathcal{D}}[h(\mathbf{x})] \right| \leq \epsilon \tag{12}$$

*for all $h \in \mathcal{H}$ with probability at least $1 - \delta$, for some universal constant $c_0$.*

Using results from (Dinitz et al., 2021), it is known that the pseudo-dimension of $\mathcal{H}$ is given by $d_{\mathcal{H}} = O(T \log T)$. With this, we are ready to prove Theorem B.1.

Our learning algorithm works as follows. For fixed $\delta, \epsilon > 0$, we choose $s$ according to Lemma B.3 so that the error rate is $\epsilon' = \frac{\epsilon}{2}$, and sample $\boldsymbol{\sigma}_1, ..., \boldsymbol{\sigma}_s \sim \mathcal{D}$ independently. Observe that functions $h \in \mathcal{H}$ map within the range $[0, \beta]$, because non-negativity is a dual constraint, and no dual variable can exceed $\beta$ due to the dual constraints. As additionally $d_{\mathcal{H}} = O(n \log n)$, the sampling complexity follows.

For each $\boldsymbol{\sigma}_i$, we calculate $\mathbf{y}^*(\boldsymbol{\sigma}_i)$ using the optimal dual algorithm. We then choose $\hat{\mathbf{y}} = \arg\min_{\mathbf{y}} \sum_{i=1}^{s} L(\mathbf{y}, \boldsymbol{\sigma}_i)$. As we use the $L_1$ norm, this can be done by choosing the median value for each coordinate, which can clearly be done in time polynomial in $s$ and $T$. Finally, with probability at least $1 - \delta$ we observe that

$$\mathbb{E}_{\boldsymbol{\sigma}\sim\mathcal{D}}[L(\hat{\mathbf{y}},\boldsymbol{\sigma})] \leq \mathbb{E}_{\boldsymbol{\sigma}\sim\mathcal{D}}[L(\hat{\mathbf{y}},\boldsymbol{\sigma})] - \frac{1}{s}\sum_{i=1}^{s}L(\hat{\mathbf{y}},\boldsymbol{\sigma}) + \frac{1}{s}\sum_{i=1}^{s}L(\hat{\mathbf{y}}^*,\boldsymbol{\sigma})$$

$$\leq \frac{\epsilon}{2} + \mathbb{E}_{\boldsymbol{\sigma}\sim\mathcal{D}}[L(\hat{\mathbf{y}}^*,\boldsymbol{\sigma})] + \frac{\epsilon}{2} \tag{13}$$

$$= \mathbb{E}_{\boldsymbol{\sigma}\sim\mathcal{D}}[L(\hat{\mathbf{y}}^*,\boldsymbol{\sigma})] + \epsilon$$

$\square$

### B.3. Culminating Result

In this section, we combine the competitiveness result, the stability result, and the learnability result into one ultimate theorem.

**Theorem B.4.** *Let $\mathcal{D}$ be a distribution, where online sequences $\boldsymbol{\sigma}\sim\mathcal{D}$ are drawn independently, and let $\mathrm{Var}(\boldsymbol{\sigma})$ denote the total variance of $\boldsymbol{\sigma}$, i.e., the trace of the covariance matrix. There is a polynomial-time learning algorithm that, given $O(|\mathcal{U}|\log|\mathcal{U}|)$ samples from $\mathcal{D}$, returns a dual laminar set cover prediction $\hat{\mathbf{y}}$ such that $\hat{\mathbf{y}}$ can be given to a learning-augmented algorithm, satisfying*

$$\mathbb{E}_{\boldsymbol{\sigma}\sim\mathcal{D}}[\mathrm{ALG}] = O\left(\mathbb{E}_{\boldsymbol{\sigma}\sim\mathcal{D}}[\mathrm{OPT}] + \mathcal{R}\cdot\beta\,\mathrm{Var}(\boldsymbol{\sigma})\right)$$

*where $\beta := \max_{e\in\mathcal{U}}\min_{S:e\in S}c_S$ and $\mathcal{R}$ is the competitive ratio for online laminar set cover.*

*Proof.* One way to phrase Theorem 2.3 is that we have the bound

$$\mathbb{E}[\mathrm{ALG}] = O\left(\mathrm{OPT} + \mathcal{R}\cdot L(\hat{\mathbf{y}},\boldsymbol{\sigma})\right) \tag{14}$$

for any instance $\boldsymbol{\sigma}$, where the expectation is taken over the randomization of $\mathrm{ALG}$. By additionally randomizing over the $\boldsymbol{\sigma}\sim\mathcal{D}$, we obtain

$$\mathbb{E}_{\boldsymbol{\sigma}\sim\mathcal{D}}[\mathrm{ALG}] = O\left(\mathbb{E}_{\boldsymbol{\sigma}\sim\mathcal{D}}[\mathrm{OPT}] + \mathcal{R}\cdot\mathbb{E}_{\boldsymbol{\sigma}\sim\mathcal{D}}[L(\hat{\mathbf{y}},\boldsymbol{\sigma})]\right) \tag{15}$$

By Theorem B.1, there is a learning algorithm which can achieve a dual laminar prediction $\hat{\mathbf{y}}$, which with probability $1-\delta$ can achieve $\mathbb{E}_{\boldsymbol{\sigma}\sim\mathcal{D}}[L(\hat{\mathbf{y}},\boldsymbol{\sigma})] \leq \mathbb{E}_{\boldsymbol{\sigma}\sim\mathcal{D}}[L(\hat{\mathbf{y}}^*,\boldsymbol{\sigma})] + \epsilon$ with the desired sampling complexity. Therefore, if we use $\hat{\mathbf{y}}$ as the dual prediction in Theorem B.1, we obtain a bound

$$\mathbb{E}_{\boldsymbol{\sigma}\sim\mathcal{D}}[\mathrm{ALG}] = O\left(\mathbb{E}_{\boldsymbol{\sigma}\sim\mathcal{D}}[\mathrm{OPT}] + \mathcal{R}\left(\mathbb{E}_{\boldsymbol{\sigma}\sim\mathcal{D}}[L(\hat{\mathbf{y}}^*,\boldsymbol{\sigma})] + \epsilon\right)\right) \tag{16}$$

We now seek to bound $\mathbb{E}_{\boldsymbol{\sigma}\sim\mathcal{D}}[L(\hat{\mathbf{y}}^*,\boldsymbol{\sigma})]$ in terms of something more common. Firstly, observe that for fixed $x$ and random variable $Y$, by convexity of $\|\cdot\|_1$ Jensen's inequality yields

$$\|x - \mathbb{E}_Y[Y]\|_1 = \|\mathbb{E}_Y[x - Y]\|_1 \leq \mathbb{E}_Y[\|x - Y\|_1] \tag{17}$$

Recall that $\hat{\mathbf{y}}^*$ is defined as the vector minimising $\mathbf{y}\mapsto\mathbb{E}_{\boldsymbol{\sigma}\sim\mathcal{D}}[L(\mathbf{y},\boldsymbol{\sigma})]$. Thus we get

$$\begin{aligned}
\mathbb{E}_{\boldsymbol{\sigma}_1\sim\mathcal{D}}[L(\hat{\mathbf{y}}^*,\boldsymbol{\sigma}_1)] &\leq \mathbb{E}_{\boldsymbol{\sigma}_1\sim\mathcal{D}}[L(\mathbb{E}_{\boldsymbol{\sigma}_2\sim\mathcal{D}}[\mathbf{y}^*(\boldsymbol{\sigma}_2)],\boldsymbol{\sigma}_1)] \\
&= \mathbb{E}_{\boldsymbol{\sigma}_1\sim\mathcal{D}}[\|\mathbf{y}^*(\boldsymbol{\sigma}_1) - \mathbb{E}_{\boldsymbol{\sigma}_2\sim\mathcal{D}}[\mathbf{y}^*(\boldsymbol{\sigma}_2)]\|_1] \\
&\leq \mathbb{E}_{\boldsymbol{\sigma}_1\sim\mathcal{D}}\mathbb{E}_{\boldsymbol{\sigma}_2\sim\mathcal{D}}[\|\mathbf{y}^*(\boldsymbol{\sigma}_1) - \mathbf{y}^*(\boldsymbol{\sigma}_2)\|_1] \quad\text{By Ineq. (17)} \\
&\leq 2\beta\cdot\mathbb{E}_{\boldsymbol{\sigma}_1\sim\mathcal{D}}\mathbb{E}_{\boldsymbol{\sigma}_2\sim\mathcal{D}}[\|\boldsymbol{\sigma}_1 - \boldsymbol{\sigma}_2\|_1] \quad\text{By Lemma 2.7}
\end{aligned} \tag{18}$$

Let's now consider $\mathbb{E}_{\boldsymbol{\sigma}_1\sim\mathcal{D}}\mathbb{E}_{\boldsymbol{\sigma}_2\sim\mathcal{D}}[\|\boldsymbol{\sigma}_1 - \boldsymbol{\sigma}_2\|_1] = \sum_{t=1}^{T}\mathbb{E}_{\boldsymbol{\sigma}_1\sim\mathcal{D}}\mathbb{E}_{\boldsymbol{\sigma}_2\sim\mathcal{D}}[|\sigma_{1t} - \sigma_{2t}|]$, where $\sigma_{it}$ denotes the $t$th component of $\boldsymbol{\sigma}_i$. Observe that because $\sigma_{it}\in\{0,1\}$, we have $|\sigma_{1t} - \sigma_{2t}|$ equal to 1 if and only if $\sigma_{1t}\neq\sigma_{2t}$. Let $p_t = \mathbb{P}_{\boldsymbol{\sigma}\sim\mathcal{D}}[\sigma_t = 1]$.

Then, the probability that $\sigma_{1t} \neq \sigma_{2t}$ equals $2p_t(1 - p_t)$, because either $\sigma_{1t} = 1$ and $\sigma_{2t} = 0$, or $\sigma_{1t} = 0$ and $\sigma_{2t} = 1$. Therefore $\mathbb{E}_{\boldsymbol{\sigma}_1 \sim \mathcal{D}}\mathbb{E}_{\boldsymbol{\sigma}_2 \sim \mathcal{D}}[|\sigma_{1t} - \sigma_{2t}|] = 2p_t(1 - p_t)$. Observe that because $\text{Var}_{\boldsymbol{\sigma} \sim \mathcal{D}}[\sigma_t] = p_t(1 - p_t)$, we have

$$\mathbb{E}_{\boldsymbol{\sigma}_1 \sim \mathcal{D}}\mathbb{E}_{\boldsymbol{\sigma}_2 \sim \mathcal{D}}[\|\boldsymbol{\sigma}_1 - \boldsymbol{\sigma}_2\|_1] = 2\sum_{t=1}^{T} \text{Var}_{\boldsymbol{\sigma} \sim \mathcal{D}}[\sigma_t] = 2\,\text{Var}_{\boldsymbol{\sigma} \sim \mathcal{D}}[\boldsymbol{\sigma}].$$

Therefore, by (18), we ultimately find the bound of $\mathbb{E}_{\boldsymbol{\sigma}_1 \sim \mathcal{D}}[L(\hat{\mathbf{y}}^*, \boldsymbol{\sigma}_1)] \leq 4\beta\,\text{Var}[\boldsymbol{\sigma}]$, which therefore implies with probability $1 - \delta$ that

$$\mathbb{E}_{\boldsymbol{\sigma} \sim \mathcal{D}}[\text{ALG}] = O\left(\mathbb{E}_{\boldsymbol{\sigma} \sim \mathcal{D}}[\text{OPT}] + \mathcal{R}\left(\beta\,\text{Var}[\boldsymbol{\sigma}] + \epsilon\right)\right) \tag{19}$$

By choosing $\delta = 1/\mathcal{R}$, the remaining case with probability $\delta$ adds a negligible cost of $\leq \mathbb{E}_{\boldsymbol{\sigma} \sim \mathcal{D}}[\text{OPT}]$ to the expected cost $\mathbb{E}_{\boldsymbol{\sigma} \sim \mathcal{D}}[\text{ALG}]$. This also adds a cost of $\ln \mathcal{R}$ to the sampling complexity, but this is dominated by $|\mathcal{U}| \ln|\mathcal{U}|$, as for any interesting case, we have $|\mathcal{U}| > \mathcal{R}$. $\qquad\square$

### B.4. Robustness

We remark that the above results provide no helpful guarantees when the error $\eta$ is large. An often desirable property is for learning-augmented algorithms to be *robust* to large prediction errors, in the sense that the performance is no worse than classical online algorithms. For Laminar Set Cover, this can be achieved by the following Lemma.

**Lemma B.5.** *Suppose we have $m$ algorithms, $A_1, ..., A_m$ for laminar set cover, each with competitive ratios $\alpha_1, ..., \alpha_m$. Then there exists an algorithm with the $A_i$ as input and ALG denoting its objective value, which for any input sequence satisfies*

$$\text{ALG} \leq m \cdot \min_{1 \leq i \leq m} \{cost(A_i(I))\} \tag{20}$$

*and therefore ALG is $m \cdot \min_{1 \leq i \leq m}\{\alpha_i\}$-competitive. If each $A_i$ is deterministic, then so is ALG.*

---

**Algorithm 4** Deterministic laminar set cover combination algorithm

---

*Proof.*   1: Initialise online algorithms $A_1, ..., A_m$ for laminar set cover.
  2: **for** times $t = 1, ..., T$ **do**
  3:     Let $\mathbf{x}_t^i$ be the online solution produced by algorithm $i$ at time $t$.
  4:     Follow an algorithm $i$ whose $\mathbf{x}_t^i$ has minimum objective value.
  5: **end for**

---

Algorithm 4 fulfills the requirements of the lemma. For a particular instance $I$, without loss of generality, assume that $A_1$ is the final algorithm which ALG follows. Therefore, $\text{cost}(A_1(I)) = \min_{1 \leq i \leq m}\{\text{cost}(A_i(I))\}$. Let $i_t$ denote the index of the algorithm that ALG follows on day $t$, and let $\text{ALG}_t$ denote the cost incurred by the algorithm on day $t$. The main inequality to observe is that, for any $1 \leq i \leq m$, we have

$$\sum_{t:i_t=i} \text{ALG}_t \leq \text{cost}(A_1(I)) \tag{21}$$

This follows, because by definition ALG could only follow $A_i$ on days $t$ for which its accumulated cost is at most that of $A_1$ at that time, and because the costs are non-decreasing throughout time, that cost is at most $\text{cost}(A_1)$. Therefore, we have the chain of inequalities

$$\begin{aligned}
\text{ALG} &= \sum_{t=1}^{T} \text{ALG}_t \\
&= \sum_{i=1}^{m} \sum_{t:i_t=i} \text{ALG}_t \\
&\leq \sum_{i=1}^{m} \text{cost}(A_1(I)) \\
&= m \cdot \min_{1 \leq i \leq m} \{\text{cost}(A_i(I)). \quad\square
\end{aligned} \tag{22}$$

# C. Omitted Results for MTS

## C.1. Learnability

In this section we establish that, under our error definition, our prediction class for MTS admits learnability with polynomial sample complexity. We fix a metric space $(M, d)$ and a time horizon $T$. We assume that the cost functions $c_1, \ldots, c_T$ are drawn from a joint probability distribution $\mathcal{D}$, which is not known to us. We want to learn

$$\widehat{w} \in W := \{(w_1, \ldots, w_T) \mid w_t \in \mathbb{R}^M \text{ and } w_t \text{ is 1-Lipschitz, for } t = 1, \ldots, T\}$$

such that, when our algorithm is provided with prediction $\widehat{w}$, its expected cost on an instance drawn from $\mathcal{D}$ is minimized. By writing $\eta(\widehat{w}; c)$ for the error of prediction $\widehat{w}$ on instance $c$, we have $\text{ALG} \leq \text{OPT} + \eta(\widehat{w}; c)$ by Theorem 3.1, and therefore we seek to find $\widehat{w}$ which minimizes $\mathbb{E}_{c \sim \mathcal{D}}[\eta(\widehat{w}; c)]$.

**Theorem C.1.** *Let $\mathcal{F} := \{c \mapsto \eta(w; c) \mid w \in W\}$, and let $n = |M|$ and $D$ be the diameter of $M$. Then, $\mathcal{F}$ is learnable with polynomial sample complexity: for a suitable value of $m$, polynomial in $n$, $T$, $B$, $1/\epsilon$, and $\ln(1/\delta)$, it holds that for every $\epsilon > 0$ and $\delta \in (0, 1)$, if we draw $m$ i.i.d. samples $c^1, \ldots, c^m \sim \mathcal{D}$, then with probability at least $1 - \delta$ every empirical risk minimizer*

$$\widehat{w} \in \arg\min_{w \in W} R_S(w), \qquad R_S(w) := \frac{1}{m} \sum_{i=1}^{m} \eta(w; c^i),$$

*satisfies*

$$\mathbb{E}_{c \sim \mathcal{D}}[\eta(\widehat{w}; c)] \leq \inf_{w \in W} \mathbb{E}_{c \sim \mathcal{D}}[\eta(w; c)] + \epsilon.$$

*Proof.* Let $S = (c^1, \ldots, c^m)$ be i.i.d. samples from $\mathcal{D}$, and let $\widehat{w}$ be any empirical risk minimizer over $W$. By Lemma C.2 we have $\text{Pdim}(\mathcal{F}) = O(n^3 T^2)$. Moreover, for $w \in W$, $\eta(w; c)$ is bounded as follows:

$$\eta(w; c) = \sum_{t=1}^{T} \|B^c(w_t) - w_{t-1}\|_{\text{sp}}.$$
$$\leq \sum_{t=1}^{T} \|B^c(w_t)\|_{\text{sp}} + \|w_{t-1}\|_{\text{sp}}$$
$$\leq 2DT.$$

In the last step, we used the definition of the Bellman operator and the fact that $w_{t-1}$ is 1-Lipschitz. Applying Lemma B.3 yields that for

$$m \geq c_0 \left(\frac{2DT}{\epsilon}\right)^2 \left(\text{Pdim}(\mathcal{F}) + \ln(1/\delta)\right),$$

with probability at least $1 - \delta$,

$$\sup_{w \in \mathcal{W}} \left|\mathbb{E}_{c \sim \mathcal{D}}[\eta(w; c)] - R_S(w)\right| \leq \epsilon/2.$$

In the event that this bound holds, let $w^\star \in \arg\min_{w \in \mathcal{W}} \mathbb{E}[\eta(w; c)]$. Using the optimality of $\widehat{w}$ and the uniform deviation bound twice, we get

$$\mathbb{E}[\eta(\widehat{w}; c)] \leq R_S(\widehat{w}) + \epsilon/2 \leq R_S(w^\star) + \epsilon/2 \leq \mathbb{E}[\eta(w^\star; c)] + \epsilon.$$

$\square$

**Lemma C.2.** $\text{Pdim}(\mathcal{F}) = O(n^3 T^2)$.

*Proof.* We equivalently seek to bound the VC-dimension of the class $\mathcal{H} := \{(c, y) \mapsto \mathbf{1}_{\eta(w; c) > y} \mid w \in W\}$. To this end, we rely on Theorem 2.3 in (Goldberg & Jerrum, 1993). As a prerequisite, we consider a membership testing algorithm which, on input $(w, c, y)$, outputs 1 if $\eta(w; c) > y$ and 0 otherwise. Recall that

$$\eta(w; c) = \sum_{t=1}^{T} \|B^c(w_t) - w_{t-1}\|_{\text{sp}}.$$

The algorithm performs the computations in the straightforward way, and runs in time $\tau = O(T \cdot n^2)$. Also, a concept in the class $\mathcal{H}$ can be represented by $T \cdot n$ real numbers. Therefore, by Theorem 2.3 in (Goldberg & Jerrum, 1993),

$$\text{Pdim}(\mathcal{F}) = \text{VCdim}(\mathcal{H}) = O((Tn) \cdot (Tn^2)) = O(n^3T^2). \qquad \square$$

### C.2. Robustness

We can make our learning-augmented algorithm robust by combining it with a classical online algorithm for MTS, by making use of the following theorem.

**Theorem C.3** (Antoniadis et al. (2023c), generalization of Fiat et al. (1994)). *Given algorithms $A_0, \ldots, A_{m-1}$ for an MTS problem, there exists an algorithm which achieves cost at most*

$$O(m \cdot \min_{i=0}^{m-1} \text{cost}_I(A_i))$$

*for any input sequence $I$. Moreover, if $A_0, \ldots, A_{m-1}$ are deterministic, then the resulting algorithm is also deterministic.*

