# OpenReview forum: "Learning-Augmented Online Minimization with Dual Predictions"
_ICML.cc/2026/Conference — ICML 2026 regular_

### Official Review · Reviewer_oiRv · 2026-02-20

**Soundness:** 3
**Presentation:** 2
**Significance:** 3
**Originality:** 3
**Overall Recommendation:** 4
**Confidence:** 4

**Summary:**

The paper studies learning-augmented algorithms for online optimization problems.
Learning-augmented algorithms (aka algorithms with predictions) are a recently popular
framework for beyond worst-case analysis, and is an established subfield in conferences such as ICML.
The authors consider two fundamental problems: online (laminar) set cover and metrical task systems.
In online set cover, a laminar set family over a known universe is given. Then elements arrive one by one and must be covered immediately via purchasing sets.
In metrical task systems, we are given a metric space and a server at the origin. At each time step, we receive a cost function over the points. Then, we can move the server to a new point (or do not move), pay the distance from the old location and the cost of the new location. The goal is to minimize the total cost.
Both are notoriously hard online problems: no constant competitive ratio can be achieved in the classical online setting. Both problems have previously been studied (extensively!) in the context of learning-augmented algorithms.

The authors present learning-augmented algorithms that achieve good (constant) competitive ratios.
if the predictions are reasonably accurate. In particular, they study the power of dual predictions,
which predict the optimal dual solution of natural LP relaxations of the considered problems.
For both problems, they design algorithms and analyze their performance. In particular, they show that
dual predictions are "stable" in the sense that if problem instances differ slightly, the optimal dual
solution also only differs slightly. Hence, predictions should achieve consistent performance if the
instance only changes slightly.
Moreover, they give sample complexity bounds for learning those predictions.

**Compliance With Llm Reviewing Policy:**

Affirmed.

**Final Justification:**

The paper shows the power of predictions of dual values for multiple problems and gives learning-augmented algorithms for it.

The rebuttal clarified my concerns and also spotted that I missed a crucial contribution.

I think the paper is a good addition to the area of learning-augmented algorithms, and should have a chance of being accepted at ICML.

**Key Questions For Authors:**

See weaknesses.

How does stability relate to the concept of "brittleness" in learning-augmented algorithms.

**Limitations:**

yes

**Strengths And Weaknesses:**

Strengths:
- The discussion about stability is interesting and uncommon.
- The paper provides new prediction models for important online problems.

Weaknesses:
- The authors do not show robustness bounds, which bound the competitive ratio independently of the prediction error. Such bounds are a key element of learning-augmented algorithms.
- The set cover results only holds in the special laminar case. Given that there exist learning-augmented algorithms for the general case, which reduces the significance of the result. There is also no outlook paragraph or discussion what fails for the general case.
- The main story of the paper builds on the fact that dual predictions are stable. However, there are no counterexamples or arguments on why existing prediction models for those problems are not stable. I do not see the benefit of these results compared to the already existing algorithms.
- Generally, there should be a stronger discussion about existing learning-augmented results and how this paper compares to those.
- There should be more information about the competitive ratio achievable for both problems before the results are stated, because the "usefulness" of predictions always depends on what is achievable without predictions.

---

> ### Author Rebuttal · Authors · 2026-03-30
>
> We thank the reviewer for their feedback. We wish to provide some clarifications below.
>
> **“authors do not show robustness”**:
> We show robustness. This is mentioned in the penultimate paragraph of “Our Results”, with more details in Section A.4 and Section C. It follows from standard techniques by which two algorithms (e.g. one following predictions, one ignoring predictions) can be combined into a hybrid algorithm whose cost is as good as the best of them up to a constant factor.
>
> **“The set cover results only holds in the special laminar case”**:
> We believe that, as we will explain, this criticism sets an unreasonable standard. Many papers study specific problems that are implicitly special cases of set cover without being criticized for not addressing set cover. The difference here is that the name “laminar set cover” makes the relationship to set cover especially explicit, so we understand the reviewer’s instinct. If our paper were only about the parking permit problem (PPP), it would presumably not be faulted for failing to handle set cover. We obtained our results for PPP first, then noticed that the techniques generalize to the broader class of laminar set cover (and MTS), so we strengthened our results accordingly. It would seem backwards to treat this added generality as a liability rather than a merit. To avoid any misunderstanding, we will revise the introduction to make this positioning clearer.
>
> **Counterexamples to stability**:
> We agree that providing counterexamples will help to clarify the contribution of our paper and will add some to the final version of our paper. We appreciate the suggestion. For laminar set cover with action predictions, an example is an instance with set $\\{1,2,\dots,n\\}$ for cost $n-\frac{1}{2}$ and each singleton set for cost $1$. If all elements arrive, the optimal solution buys the large set; if one arrival is missing, the optimal solution instead buys $n-1$ singletons. For MTS, there are various counterexamples to action predictions. For example, consider the 3-server problem on the line with all servers starting at 0, and the request sequence repeats -2, -1, 2, 1 many times and ends with a final request at -2. Then the unique optimal solution keeps two servers stationary at -2 and -1 and the third server goes back and forth between 1 and 2. Changing the final request to 2 completely changes/mirrors the unique optimal solution, with two servers stationary at 1 and 2 and the third going back and forth between -2 and -1.
> Instability not only complicates the learning task, it can also lead to disastrous performance of learning-augmented algorithms even when the anticipated future differs from the truth in a single request (and even when algorithms have smoothness guarantees): We give a concrete example for the well-known next-request time prediction model for paging in our response to Reviewer 2h9U.
>
> **Needs “stronger discussion about existing learning-augmented results and how this paper compares”**:
> Most comparison to prior work takes place in the “What to predict?” and “Related Work” sections, and we note that reviewer kiP4 praised our literature review as “very comprehensive” and “like a mini-survey”. Nonetheless, we believe the present reviewer’s concerns are addressed in our response to Reviewer 2h9U. If not, and if there is something specific that seems missing, please let us know and we’ll be happy to provide more details.
>
> **“more information about the competitive ratio achievable for both problems before the results are stated”**:
> We will move this information earlier to make it clearer. For general MTS, when $n$ is the number of points in the metric, it is $O(\log^2 n)$ randomized and $2n-1$ deterministic (currently discussed in Related Work). But this is only part of the story as there are many special cases of MTS with better ratios, and it seems impossible to discuss them all. For example, $k$-server is $O(k)$-competitive, convex body chasing is $O(\text{dimension})$-competitive, width-$w$-layered graph traversal is $O(w^2)$-competitive etc. We improve all of these ratios to $O(1)$ if predictions are good enough (and justification of why “good enough” is achievable is via our learnability and stability results, and experiments). For laminar set cover it is also diverse. For PPP, it is $\Theta(\log K)$ randomized and $\Theta(K)$ deterministic, where $K$ is the number of permit types (we currently discuss this only in the context of our experiments and agree this is much too late). The same ratio is also tight for general laminar set cover (when $K$ is the maximal number of sets in which an item can appear). We will make sure that some discussion about these results appears earlier in the final version of our paper.
>
> **Brittleness vs stability**: The term brittleness has been used as meaning the opposite of smoothness. Smoothness and stability are different concepts, though part of their motivation is similar (cf. response to Reviewer 2h9U).

---

> > ### Author Rebuttal · Reviewer_oiRv · 2026-03-31
> >
> > I thank the authors for their detailed response.
> >
> > - I admit that I oversaw the paragraph on robustness. I was mainly focused on the main theorems given their "property-list" type style, where I would have suspected that robustness is a bulletpoint. Perhaps you may think about adding it there to make it more visible. This definitely improves my assessment of the paper. I am again sorry for missing this part.
> >
> > - Given that you categorize my critique as "unreasonable standard", I was a bit surprised that you are not giving concrete examples and/or strong references of laminar set cover problems that clearly support your claim. Can you give more details here? In any case, I understand your reasoning and your very much appreciate a better positioning in the introduction.
> >
> > - Since the field of learning-augmented algorithms is very big, I am still not convinced that "praised" is the right word here. In any case I appreciate your effort of adding details.
> >
> > - Thank you for adding details on earlier results. This will improve the paper.
> >
> > In summary, I am now convinced that the paper should have a chance of being accepted, and I will adjust my score accordingly.

---

> > > ### Author Response · Authors · 2026-04-07
> > >
> > > We thank the reviewer for their response, and we are glad that our clarifications were helpful.
> > >
> > > We hope that our comment about “unreasonable standard” did not come across as insulting (and we apologize in case it did); as we said, we understand where the concern about laminar set cover is coming from, given the problem’s name and definition directly emphasize the link to set cover. What we mean is: Expecting results on a problem (or problem class) to extend to the most general superclass would not seem to be the right standard, and is not the standard that has been applied to other special cases of set cover before. We’re not sure if we understand which claim exactly the reviewer would like us to support with references; there are several papers specific to, say, ski rental in the learning-augmented literature (many listed at https://algorithms-with-predictions.github.io/), and these papers are not typically faulted for their lack of handling set cover even though set cover is a generalization of ski rental.
> > >
> > > Some problems that can be modeled as *laminar* set cover are:
> > > - Ski rental: A singleton set for each day (corresponding to renting on that day) and a set containing all days (corresponding to buying). Given that ski rental admits constant competitive algorithms, our results might seem vacuous for it due to the O-notation in Theorem 1.1(a). But the bound in Theorem 1.1(a) can actually be stated more precisely as $(1+\epsilon)\text{OPT}+O(\frac{\mathcal R \eta}{\epsilon})$ by choosing $\alpha=1/(1+\epsilon)$ in our algorithm (see Theorem 2.3). We shall update Theorem 1.1(a) to make this clearer.
> > > - Dynamic power management, which is essentially equivalent to solving multiple ski rental instances in a row (cf. Antoniadis et al., 2021).
> > > - Parking permit problem (PPP): Modelling it as laminar set cover loses only a constant factor (Meyerson, 2005), and it’s a good example showing how a seemingly non-laminar problem can be cast as laminar set cover. The reduction rounds permit durations to powers of 2, and rounds the start and end date of permits so that a permit of duration $2^i$ begins on a day $j\cdot 2^i+1$ and ends on a day $(j+1)2^i$ for some integer $j$.
> > > - Multi parking permit problem, which is the generalization where on day $t$ there is a demand $d_t\in\mathbb Z_{\ge 0}$ and one needs to own at least $d_t$ permits. The problem is known to reduce to PPP [1], and thus to laminar set cover.
> > > - Steiner leasing problem and related problems discussed in [1], which also reduce to PPP.
> > > - Sum-radii clustering in ultrametrics (a.k.a. HSTs) [2]: points arrive online and must be assigned to clusters. Each cluster incurs a fixed opening cost plus a cost equal to its radius. The ultrametric/HST structure means clusters are subtrees, hence the corresponding sets form a laminar family. The HST case is specifically highlighted in [2], and superconstant competitiveness in the classical online setting holds even on HSTs. Since laminar families can be arranged in trees, laminarity is natural to come up in the context of tree/HST embeddings, which are central to many online algorithms problems (as arbitrary metrics can be approximated by trees).
> > >
> > > We will expand the discussion of examples of laminar set cover in the final version of our paper.
> > >
> > > For general set cover, at least when using its standard LP formulation, there are worst-case instances that remain hard even when an optimal dual solution is known upfront. This is why our results do not easily extend. We will add an example of this to the appendix of the final version of our paper. But it is an interesting question whether a different way of modelling set cover might lead to more useful duals (e.g., just like for MTS, our results rely on our particular LP formulation, whereas a very similar LP formulation for MTS would have been unsuitable; cf. lines 288-292). Deriving results similar to ours for problems that do not fall into either MTS or laminar set cover is an interesting future direction, and we will add some outlook discussing this, as suggested in the original review.
> > >
> > > [1] M. S. de Lima, M. San Felice, O. Lee: On Generalizations of the Parking Permit Problem and Network Leasing Problems. 2017
> > >
> > > [2] D. Fotakis, P. Koutris: Online Sum-Radii Clustering. 2012

---

### Official Review · Reviewer_kiP4 · 2026-03-04

**Soundness:** 4
**Presentation:** 4
**Significance:** 3
**Originality:** 3
**Overall Recommendation:** 5
**Confidence:** 5

**Summary:**

The paper studied learning-augmented algorithms for online optimization with predictions of the variables of the *dual linear program*. In particular, the paper considered two online minimization problems: laminar set cover and metrical task systems. The two problems are sufficiently versatile to solve an array of optimization problems in machine learning. Furthermore, a common feature for these problems is that they can be written into LPs, and we can easily find their duals. The core algorithms in the paper utilize the predictions of the dual variable to improve the approximation guarantees. For predictions with error $\eta$, the paper gives the following algorithmic results:

- For laminar set cover, a framework that given an $R$-approximation algorithm for the online problem (without prediction) and the predictions, return an $(O(1), R\cdot \eta)$-approximation. Notably, the multiplicative approximation becomes $O(1)$, and $R$ only appears on the *additive error* term.
- For metrical task systems, an algorithm that given a prediction of the dual variable $w_t(s)$, at every time, computes a $(1, \eta)$-approximation.

In addition to the approximation guarantees, the paper also provided “stability” guarantees, meaning that if the prediction is, e.g., for a “close” instance, the prediction error is bounded by the difference between the instances.

**Main techniques.** The techniques used in the algorithms are based on structural observations of the dual linear programs. For the laminar set cover problem, the key observation appears to be that once a dual variable is “fractionally packed” by an $\alpha$ factor, the error induced by picking the set scales with $1/\alpha$, and the prediction error will be a linear additive error. For the MTS problem, the target $w_t(s)$ function captures a reverse-time notion for the costs, which leads to the improved bounds.

**Compliance With Llm Reviewing Policy:**

Affirmed.

**Final Justification:**

I believe all of my concerns have been resolved. For the experiment, I guess one of the reasons for me to comment about "not significant" is because I think the difference in the competitive ratio for <10 constant vs. 1.x is not that much. But anyway, I appreciate the papers contributions.

I also understand that I'm the most supportive reviewer, and I'll support the acceptance for paper to a reasonable extent.

**Key Questions For Authors:**

- For the laminar condition of the online set cover, do you need it in the proof of Lemma 2.4? If not, it seems the condition would only be used in the stability proof.
- You explained why using $\|\cdot\|_{sp}$ norm (intead of $\ell_\infty$) is OK around line 318. However, in the analysis, what makes the $\|\cdot\|_{sp}$ norm work and the $\ell_\infty$ norm not work?

**Limitations:**

Yes (no foreseeable negative societal impact)

**Strengths And Weaknesses:**

**Strength.**
- Well-justified motivation: I have to admit that upon first reading the paper, my perception was quite negative since “predicting dual variables” seems to be something impossible or at least not straightforward. However, upon reading the authors’ justifications, it seems getting such predictions is quite possible, although we cannot simply use an “off-the-shelf” ML model. This part might get some controversy during the review/rebuttal process. Anyway, I think I would stand **in favor of such a model** due to the fact that it is actually possible for learning to happen.
- The paper is technically quite sound. I did not get time to carefully check all the proofs and details. However, a spot check makes me believe in the general strategy of the algorithms.
- The paper is also well-written (to the extent that it convinced me to “flip” my first impression). :)
- The literature review seems to be very comprehensive and is like a mini-survey.

**Weakness**:
- The perceived controversy of the motivation: as I said, I can see people will have divergent opinions about your model that predicts dual variables, and I would completely understand potential criticisms. Among other reasons, this is not quite intuitive for the general ML audiences, and we cannot easily use off-the-shelf ML models. You should be prepared to answer these issues.
- The techniques used in the two algorithms are quite “ad-hoc”, meaning that it is hard to extract some general mechanisms to apply to other problems with dual parameter predictions.
- In the experiments (for both parking permit and $k$-server problems), the empirical performance of the learning-augmented algorithm is not so much better than a simple randomized algorithm.

Based on the merits and criticisms, my overall opinion of the paper is quite positive, and I’ll be happy to see the paper get accepted.

---

> ### Author Rebuttal · Authors · 2026-03-30
>
> We thank the reviewer for their review and the positive evaluation, and address some questions/comments from the review below.
>
> **Perceived controversy about learning duals:** As the reviewer points out, these duals are learnable, and our experiments support that one can obtain high-quality predictions easily from historical data on real-world instances. In fact, this contrasts with action predictions of prior work, of which it is much less clear how they can be obtained in practice (in part due to the instability/volatility of optimal actions).
>
> **Techniques:** We agree that the precise execution of how to incorporate dual predictions may vary depending on the problem, just like it is quite different between laminar set cover and MTS. Our paper serves as a proof-of-concept: We show for two distinct classes of problems that dual predictions have several beneficial properties. These classes already capture many problems, and we expect that dual predictions will have many additional applications.
>
> **Experiments:** We are surprised by the perception that our learning-augmented algorithms are not much better than randomized algorithms in the experiments. Maybe it seems that way because in Figure 1, the gap between learning-augmented and randomized is not as large as the gap between randomized and deterministic? Notice that the competitive ratio of our learning-augmented algorithms is very close to 1 in our experiments, so it performs almost as well as the offline optimum. In contrast, the performance of all the online algorithms deteriorates as the number of permits/servers increases (as expected because classical online algorithms analysis suggests that these parameters determine the problem’s hardness). We may add a horizontal line where the competitive ratio is 1 to the figures (corresponding to offline optimum cost) to make this more apparent.
>
> **Laminar condition**: The assumption of laminarity is indeed necessary for Lemma 2.4 so that we do not make multiple charges to the same element in our charging argument.
>
> **Span seminorm**: Our results would also hold when defining error in terms of $\ell_\infty$-norm instead of span seminorm. The reason we use the span seminorm is that it strengthens the result: The span seminorm is at most twice the $\ell_\infty$-norm, so it directly implies similar performance bounds in terms of the $\ell_\infty$-error. But span seminorm can be much smaller than $\ell_\infty$-norm (this is what we mean by “more forgiving” in line 318), hence our performance bound expressed in terms of span seminorm is stronger than if it were expressed in terms of $\ell_\infty$-norm.

---

> > ### Author Rebuttal · Reviewer_kiP4 · 2026-04-04
> >
> > Thanks for the responses. I believe all of my concerns have been resolved. For the experiment, I guess one of the reasons for me to comment about "not significant" is because I think the difference in the competitive ratio for <10 constant vs. 1.x is not that much. But anyway, I appreciate the papers contributions.
> >
> > I also understand that I'm the most supportive reviewer, and I'll support the acceptance for paper to a reasonable extent.

---

### Official Review · Reviewer_2h9U · 2026-03-11

**Soundness:** 4
**Presentation:** 3
**Significance:** 2
**Originality:** 3
**Overall Recommendation:** 4
**Confidence:** 4

**Summary:**

The paper studies two classes of online minimization problems, metrical task systems and laminar set cover, in the framework of algorithms with predictions.

In both settings, requests arrive online and the goal is to satisfy all requests with minimum total cost through irrevocable decisions. In the former problem, each decision corresponds to moving a server to a location in a metric space and paying a cost consisting of the traveled distance and a service cost that depends on the request and the new server location. In the latter problem, each request is an element from a ground set. We are given a laminar family of subsets of this ground set, each with an associated cost. At every time step, the algorithm must maintain a subcollection of these sets that covers all elements that have arrived so far while minimizing the total cost.

The prediction model considered in this paper provides predictions of the optimal dual solution of the LP relaxations for these problems. The prediction error is defined separately for each problem. For both settings, the authors prove three properties:
1. Small changes in the instance lead to small changes in the dual variables, implying that the predictions are stable.
2. The performance guarantee, expressed as the competitive ratio, degrades smoothly as the prediction error increases.
3. The predictions can be learned with polynomial sample complexity.

The paper also includes experiments on one instance from each problem class, the $k$-server problem for metrical task systems and the parking permit problem for laminar set cover, using real-world data to support the theoretical guarantees.

**Compliance With Llm Reviewing Policy:**

Affirmed.

**Final Justification:**

My final recommendation is weak accept. The rebuttal addressed my main concerns and reinforced my prior assessment.

**Key Questions For Authors:**

I do not have any questions for the authors.

**Limitations:**

yes

**Strengths And Weaknesses:**

**Soundness**

The paper appears technically sound. The claims are supported by rigorous theoretical analysis as well as empirical evaluation. The prediction model and the corresponding error measures are reasonable, and the experiments appear to be designed appropriately.

**Presentation**

The paper is well written and easy to follow. In terms of positioning in the literature, the paper cites a large body of related work. However, when discussing prior work on learning-augmented algorithms for problems that can be viewed as instances of these two classes, the paper does not clearly explain how those results compare to the results obtained here.

**Significance**

The two classes of problems considered in the paper are fairly general and capture many important online minimization problems. Many specific examples within these classes have already been studied in the framework of learning-augmented algorithms. While the results in this paper apply more broadly, the lack of clear positioning relative to prior work makes the overall significance somewhat unclear to me.

The predictions considered in this paper correspond to the optimal dual solution rather than the optimal primal solution or the arrival sequence. In addition to establishing performance guarantees that depend on prediction quality, which is the main objective in most algorithms-with-predictions papers, the paper also shows that these dual predictions are stable, meaning they change only slightly when the instance changes slightly.

**Originality**

The main novelty of this work is the use of predictions on dual variables for online minimization problems. This contrasts with prior work that uses predictions on dual variables of an auxiliary problem rather than the main LP, or combines dual predictions with other types of predictions.

---

> ### Author Rebuttal · Authors · 2026-03-30
>
> We thank the reviewer for their review and positive evaluation. We appreciate the comments on clarifying the positioning relative to prior work, which we address below:
>
> As prior work on (special cases of) the problems we study uses different types of predictions, the competitive ratios achieved in these models are not directly comparable. For example, Antoniadis et al. (2023c) give a bound for MTS of the form $\text{ALG}\le O(\text{OPT}+\eta)$ that at first glance looks similar to our Theorem 1.2(a), but $\eta$ is a completely different quantity in their case, indicating the prediction error of action predictions (i.e., the predictor directly tells the algorithm what to do, and by essentially following the predicted actions their smoothness bound follows via triangle inequality). Thus, a conceptual comparison seems more important than listing competitive ratios that are incomparable across models. Also for learning-augmented set cover, prior work focuses on action predictions. The usefulness of action predictions is intuitively evident (if you get good advice of what to do, just do it), and the main focus in those works is typically how to incorporate them robustly. (We achieve robustness as well at only a constant factor loss via similar techniques, cf. penultimate paragraph in “Our Results”.) But **action predictions shift an enormous burden onto the predictor**, as it is a strong assumption that high-quality action advice is given. Action predictions are not known to be learnable, and they are highly unstable: A tiny change in the instance can completely change the optimal actions. In contrast, our framework goes beyond action predictions and justifies the existence of good predictions via learnability and stability results.
>
> A separate problem of unstable predictions in prior work can be seen in Lykouris & Vassilvitskii’s well-known model for paging (special case of MTS) with next-request time prediction. Here, instability has the effect that replacing just a *single* request in an otherwise perfectly predicted sequence can make the performance of Wei's (2020) state-of-the-art algorithm for this model deteriorate to a competitive ratio $\Omega(\log k)$, as bad as a worst-case online algorithm without predictions! We provide an example at the end of this response that exhibits such behavior. **So stability not only ensures existence of good predictions, its absence can completely eradicate all supposed benefits of predictions even under a single change in the request sequence. Remarkably, this is true even for algorithms satisfying the smoothness property** (i.e., competitive ratio degrades gracefully as a function of prediction error), which may seem surprising as smoothness is motivated by the desire to prevent such scenarios. The reason why smoothness does not prevent this is that without stability, small instance changes can cause huge prediction errors.
>
> We hope the above discussion clarifies the significance of our contribution, and we will revise our paper to better highlight the relationship to previous results. We thank the reviewer for their comments and would be happy to provide more information in case they have remaining concerns.
>
> **Example showing failure due to instability for paging.** In the model of Lykouris & Vassilvitskii for paging, each request to a page $p$ comes with a prediction of the next time when $p$ is requested again. The best learning-augmented algorithm for this setting by Wei (2020) works as follows: on a cache miss, evict the page with the furthest-in-future *predicted* next-request time, but default to a standard online algorithm ignoring predictions if it has smaller overall cost. Now consider the following request sequence:
>
> Stage 1: Request $1,2,\dots,k,2$
>
> Stage 2: Repeatedly request $2,3,\dots,k+1$ many times
>
> Stage 3: An adversarial sequence on the pages $2,3,\dots,k+2$ such that the online algorithm without predictions attains its worst-case competitive ratio ($\Omega(\log k)$ if randomized, $\Omega(k)$ if deterministic)
>
> Now imagine the *anticipated* request sequence is the same except the last request of Stage 1 is 1 instead of 2, and next-request time predictions are sampled according to this anticipated sequence.
>
> The first request, at page 1, is accompanied by a prediction that page 1 will be requested again at time $k+1$. Thus, after the first $k$ requests fill the initially empty cache, the evict-furthest-predicted policy would refuse to ever evict page 1 as it anticipates a next request to 1 earlier than any other request (although in the true request sequence page 1 is never requested again). But this would lead to cost tending to infinity during Stage 2, whereas the online algorithm without predictions would serve Stage 2 for constant cost by keeping pages $2,3,\dots,k+1$ in cache. This triggers the defaulting behavior, whereby predictions will be ignored and the overall algorithm is as bad as the online algorithm without predictions.

---

> > ### Author Rebuttal · Reviewer_2h9U · 2026-04-02
> >
> > Thank you for your thorough response. My concerns have been addressed. That said, I am maintaining my score.

---

### Decision · Program_Chairs · 2026-04-30

**Decision:**

Accept (regular)

**Comment:**

The paper proposes learning-augmented algorithms for two online problems. The novelty is that the algorithms take as predictions presumably optimal *dual* solutions of LP relaxations. (This is a new idea for these problems, but there are several known learning-augmented algorithms for other problems with predicted duals). The paper shows that: small changes in instance lead to small changes in duals, dual predictions are learnable, the proposed algorithms behave smoothly wrt the prediction error. It also includes a small experiment.

All reviewers agree that the paper is interesting, sound and well written; they all vote for acceptance.